# DropCompute: simple and more robust distributed synchronous training via compute variance reduction

**Niv Giladi**[1,2]* **Shahar Gottlieb**[1,2]* **Moran Shkolnik**[1,2] **Asaf Karnieli**[2]

**Ron Banner**[2] **Elad Hoffer**[2] **Kfir Yehuda Levy**[1] **Daniel Soudry**[1]

[1]Technion - Israel Institute of Technology
[2]Habana-Labs

{giladiniv, moranshkolnik, elad.hoffer, kfiryehud, daniel.soudry}@gmail.com
{sgottlieb, akarnieli, rbanner}@habana.ai

## Abstract

**Background.** Distributed training is essential for large scale training of deep neural networks (DNNs). The dominant methods for large scale DNN training are synchronous (e.g. *All-Reduce*), but these require waiting for all workers in each step. Thus, these methods are limited by the delays caused by straggling workers. **Results.** We study a typical scenario in which workers are straggling due to variability in compute time. We find an analytical relation between compute time properties and scalability limitations, caused by such straggling workers. With these findings, we propose a simple yet effective decentralized method to reduce the variation among workers and thus improve the robustness of synchronous training. This method can be integrated with the widely used *All-Reduce*. Our findings are validated on large-scale training tasks using 200 Gaudi Accelerators. A reference implementation[2] is provided.

## 1 Introduction

Deep Neural Networks (DNNs) training continues to scale over size and computational footprint, as a result of a higher number of trainable parameters, wider and deeper models, and growing amounts of training data. As improvements in model quality (measured by test loss, for example) (Kaplan et al., 2020) lead over hardware capabilities (Hooker, 2020), this scale-up translates into a need for a growing number of training devices working in tandem (Chowdhery et al., 2022), turning distributed training to the standard approach for training DNNs on a large scale.

Distributed training typically refers to three parallelism paradigms — data parallel, model parallel and layer pipelining (Ben-Nun & Hoefler, 2019). Several variants and hybrid solutions exist in modern implementations such as tensor parallel (Narayanan et al., 2021) and parameter sharding (Rajbhandari et al., 2020; Rasley et al., 2020). These can be used separately or combined as they are orthogonal to each other. Mainly, data parallelism is straightforward, where the data is sharded among workers, and all workers share the same global model state. At each step, workers compute gradients locally and then aggregate them before taking an optimization step. When training synchronously, workers update their parameters in lockstep. This ensures that all workers hold a consensus on the same model and that gradients are averaged over all workers before being applied to the model. This approach is easy to implement and allows for good convergence properties, and correspondingly is the prevalent optimization method.

---

*Equal contribution
[2]https://github.com/paper-submissions/dropcompute

37th Conference on Neural Information Processing Systems (NeurIPS 2023).

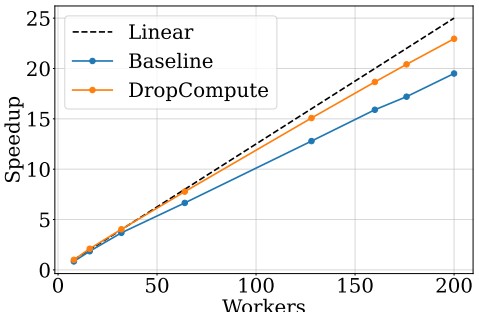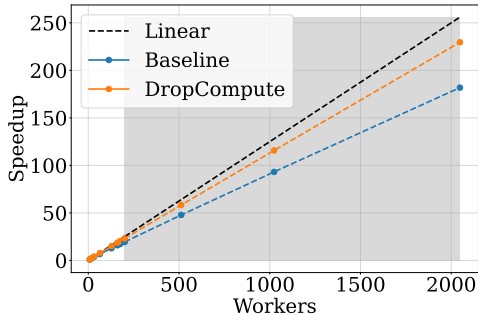

Figure 1: **DropCompute improves robustness and scalability of synchronous training.** A scale graph, showcasing the proposed method runtime performance of synchronous training of a 1.5 billion parameter model with additive noise, simulating compute variance. The baseline represents existing synchronous training and the dashed black line is linear scaling. (left) Real measurements of up to 200 workers. (right) An extrapolation to 2048 workers using a theoretical estimation, also proposed in the paper. More details are provided in section 5.2.

Although state-of-the-art models use synchronous optimization for training, synchronous methods scale poorly, as stragglers and communication overhead might severely deteriorate system utilization. Many resources are invested in alleviating these issues in large-scale training systems since even minor improvements can be worth hundreds of thousands of dollars. These issues are exacerbated as the required scale grows, even in homogeneous high-performance computing clusters (Petrini et al., 2003; Hoefler et al., 2010). In this paper, we are interested in cases where significant computing variance between the workers exists. This includes (but is not limited to) straggling workers.

For instance, certain learning tasks entail heterogeneity in the required computation of data, such as varying sentence lengths in language processing, or different image sizes and frame numbers in computer vision. In addition, recent state-of-the-art models use all three parallelism paradigms (data (DP), tensor (TP), and pipeline (PP) parallelism), thus each data parallel node is a set of processing units (accelerators) communicating between them (via TP and PP) to calculate the model gradients collectively. This could potentially intensify compute variance between data parallel workers. Moreover, sub-optimal hardware systems can also lead to straggling workers. Although some sources of heterogeneity can be mitigated by non-trivial engineering work, they still persist as challenges that can introduce compute variance and substantial performance drops. This is further discussed in appendix A.

In this paper, we suggest a simple, yet effective method called *DropCompute* to improve the robustness and scalability of synchronous optimization in the face of compute variance. We model the compute time as a random variable and show that under reasonable assumptions, a tail of straggling workers slows down the system at a rate that is not proportional to the contributed compute by these straggling workers. We harness the gradient accumulation method widely used in Large Language Models (LLMs) (Ott et al., 2018; Liu et al., 2019) to implement the method in a few lines of code on a relevant large-scale learning task.

The contributions of our work include:

- *DropCompute*: a novel, decentralized method to better handle heterogeneity or stragglers without additional hyper-parameters. *DropCompute* is hardware and framework agnostic, runs on top of existing optimizers, and can also be combined with other methods that improve other aspects of robustness such as communication overhead.

- A theoretical convergence proof of SGD with stochastic batch size, as in *DropCompute*.

- A theoretical runtime analysis on standard synchronous training and the proposed method. We find an approximation of the expected speedup using *DropCompute*, and show this speedup goes to infinity as the number of workers grows.

- An empirical evaluation of the proposed method on a relevant large scale task, using up to 200 accelerators connected with high bandwidth communication. For example, when using *DropCompute* before and after system optimizations (for both software and hardware) we show accelerations of 18% and 5%, respectively.

## 2 Related Work

The challenge of training deep neural networks on a large scale has been extensively explored. With rapidly growing models and data sizes, numerous works tackled the weaknesses in synchronous DNN training on a large scale and suggested methods to alleviate these weaknesses.

**Redundancy methods.** This line of work addresses the straggling worker problem using a redundancy mechanism. Redundant workers or redundant data are used such that straggling workers will not slow down the entire system (Chen et al., 2016; Bitar et al., 2020). These methods provide better robustness to synchronous training, even in the event of a complete failure of a subset of the workers or considerable communication slowdown. However, the robustness is limited by the redundancy factor, and more generally, more compute resources are required and full utilization cannot be achieved. In addition, some coordination method is required to keep the training synchronous, i.e., keeping consensus between the model replicas of the workers. In particular, Chen et al. (2016); Bitar et al. (2020) use a centralized approach of a parameter server to determine which workers are left out at each iteration. Modern large-scale systems use decentralized variants of *All-Reduce* (von Luxburg et al.; Patarasuk & Yuan, 2009), so it is not trivial to determine which workers should be considered at each step, given that each worker can see a different subset of straggling workers. Moreover, combining redundancy with communication primitive collectives (e.g., *All-Reduce*) requires adaptation to existing underlying frameworks (Sanders et al., 2019).

**Asynchronous optimization.** Another approach is introducing asynchrony to the optimization. Asynchronous training is inherently more scalable than synchronous training by being robust to all kinds of workers and communication faults. This includes periodic synchronization by exchanging parameters every $\tau$ optimization steps (Stich, 2019; Lin et al., 2020; Wang & Joshi, 2021; Zhang et al., 2015; Wang et al., 2020; Li et al., 2020b,a), approximate distributed averaging where each worker communicates with a subset of workers each step (Jiang et al., 2017; Lian et al., 2017; Assran et al., 2019; Yang et al., 2020), and many more. These works provide better scale-up properties and improve time performance. The main drawback is asynchronous optimization itself. In practice, the convergence is less stable, and more optimization steps are needed to generalize as well as synchronous training. In addition, hyperparameters should be chosen more precisely to guarantee convergence and generalization properties (Giladi et al., 2019; Mitliagkas et al., 2016). Due to these issues, asynchronous methods are less commonly used on a large scale.

**Sparse and compressed communication.** Alternatively, several works addressed only the communication overhead. A common technique is to reduce the amount of data exchanged by the workers at each step. This can be done by gradient pruning (Xu et al., 2021), gradient compression (Seide et al., 2014; Chen et al., 2020; Tang et al., 2021) or low-rank approximation (Vogels et al., 2019). These works reduce the communication overhead in a deterministic form while ignoring any compute variance across processing workers. This makes these works orthogonal to ours and potentially can be combined with our method.

## 3 Reducing Compute Variance

This paper proposes a method called *DropCompute* that improves the robustness of synchronous training by reducing compute variance. First, we describe the vanilla synchronous training framework. Then, we introduce the proposed approach.

### 3.1 Problem setup

We start with a model formulation for data-parallel synchronous SGD training with $N$ workers, where each worker holds a replica of the model parameters $\theta$. In parallelism paradigms that combine TP or PP with DP, $N$ represents the count of data-parallel workers, not the total number of workers involved in training. Given a dataset $\mathcal{D}$, we are interested in minimizing the empirical loss

$$\mathcal{L}(\mathcal{D}, \theta) = \frac{1}{|\mathcal{D}|} \sum_{z \in \mathcal{D}} \ell(z, \theta),$$

where $\ell(z, \theta)$ is the loss with respect to data-point $z$ and the model parameters $\theta$. At each step, the workers calculate gradients based on a local batch and then aggregate the gradients before taking an

optimization step. An equivalent strategy would be that the workers aggregate the parameters after taking an optimization step based on the local gradients. The aggregation is done in a decentralized fashion, such as *AllReduce*.

We also consider the use of gradient accumulation to increase the size of the local batch. Gradient accumulation breaks down each worker's local batch into $M$ micro-batches for computation, which enables reaching a large global batch size beyond hardware capacity. This is common practice in training of LLM on large scale where a substantial number of accumulations are being utilized (Smith et al., 2022; Nvidia, 2023). In each iteration $i$ and accumulation $m$, the gradients of the loss function with respect to the model parameters are computed, denoted by

$$g_n^{(m)}(\theta_i) = \nabla \mathcal{L}(\mathcal{D}_{i,n}^{(m)}, \theta_i),$$

where $\mathcal{D}_{i,n}^{(m)} \subseteq \mathcal{D}$ is the micro-batch $m$ of worker $n$, sampled without replacement from $\mathcal{D}$, and we assume a constant micro-batch size $|\mathcal{D}_{i,n}^m|$. The gradients accumulated by worker $n$ at step $i$ are

$$g_n(\theta_i) = \frac{1}{M} \sum_{m=1}^{M} g_n^{(m)}(\theta_i).$$

Finally, the workers aggregate and average the computed gradients to update the model parameters

$$\theta_{i+1} = \theta_i - \eta g(\theta_i); \quad g(\theta_i) = \frac{1}{N} \sum_{n=1}^{N} g_n(\theta_i), \tag{1}$$

where $\eta$ is the learning rate. Equation 1 requires all workers to receive all gradients before updating the parameters. This communication restriction is what makes the training synchronous and ensures the model parameters are the same for all workers. However, due to this restriction, the slowest worker at each step dictates the iteration time. More generally, any variation in computation time among workers will lead to workers with idle time. Therefore, to improve the efficiency and robustness of the synchronous training process, we need to reduce the computation time variance among workers (namely, the compute variance).

### 3.2   Our method: *DropCompute*

To mitigate the effect of compute variance and to improve the robustness of synchronous training, we propose a simple yet effective method called *DropCompute*. *DropCompute* reduces the compute variance by introducing a *compute threshold* after which workers have to stop calculating local gradients and move directly to the communication phase, i.e., Equation 1.

In each iteration $i$, each worker $n$ measures the time while calculating its local batch, and compares it against a given threshold $\tau$, which is set as described in section 4.4. If that time exceeds the *compute threshold*, the worker stops and sends the gradients it has calculated so far. The rest of the training remains unchanged, thus synchronous training is intact. The pseudo-code of *DropCompute* is shown in Algorithm 1. This method maintains a decentralized approach while improving its robustness to compute variance and stragglers in particular. Since our method drops samples, the batch size is no longer deterministic. Specifically, the total batch size (the total number of samples used in one iteration by all workers) can be smaller than the maximal total batch size $b_{\max} = NM|\mathcal{D}_{i,n}^{(m)}|$. Nevertheless, we show both theoretically and empirically that this makes no difference to convergence or generalization when the number of computed samples remains the same. Next, we analyze this method in section 4 and evaluate it in section 5.

## 4   Method Analysis

After establishing the notion of compute variance and the formulation of the proposed method, we analyze synchronous training and the potential value of the proposed method on the training time performance. To do so, we start with convergence guarantees when using *DropCompute*. Then, we theoretically analyze the computation time with synchronous training and when using *DropCompute*. Through this analysis, we estimate the potential speedup over vanilla synchronous training.

---
**Algorithm 1** DropCompute on worker $n$ at iteration $i$
---
1: **Input:** model parameters $\theta_i$; total number of micro-batches $M$; compute threshold $\tau$
2:     local batch data $\mathcal{D}_{i,n} = \{\mathcal{D}_{i,n}^{(1)}, \mathcal{D}_{i,n}^{(2)}, \cdots, \mathcal{D}_{i,n}^{(M)}\}$; step time $T_n$ to compute local batch $\mathcal{D}_{i,n}$
3: **Initialize** step time $T_n = 0$ and accumulated gradients $g_n(\theta_i) = 0$
4: **do (1) and (2) in parallel:**
5:     **(1) for** $m = 1, \ldots, M$ **do**
6:             $g_n^{(m)}(\theta_i) = \nabla\mathcal{L}(\mathcal{D}_{i,n}^{(m)}, \theta_i)$                                    ▷ Compute gradient
7:             $g_n(\theta_i) \leftarrow g_n(\theta_i) + g_n^{(m)}(\theta_i)/M$                          ▷ Accumulate gradients (atomic)
8:     **(2) wait for** $T_n > \tau$ **and break for loop (1)**
9: **Output:** $g_n(\theta_i)$                                                              ▷ *AllReduce*
---

## 4.1 Convergence analysis of *DropCompute*

Using *DropCompute*, the batch size is no longer fixed, but a random variable, which can also potentially depends on the data samples. To the best of our knowledge, this setting is somewhat different from existing convergence guarantees. Therefore, in this section, we provide convergence guarantees for *DropCompute* by analyzing the convergence of SGD with stochastic batch size.

**Assumption 4.1.** *Following the notations in section 3.1, consider a possibly non-convex smooth loss function $\mathcal{L}(\mathcal{D}, \theta)$, with a global minimum $\theta^* \in \mathbb{R}^d$, and the following (commonly used) assumptions*

1. **L-smooth**: *All functions $\mathcal{L}(\cdot, \theta)$ are with L-Lipschitzian gradients.*

2. **Unbiased estimation**: *Each $\nabla\ell(z, \theta_i), z \in \mathcal{D}$ is an unbiased estimate of $\nabla\mathcal{L}(\mathcal{D}, \theta_i)$ which is the true (full batch) gradient at $\theta_i$.[2] Namely, $\forall i : \mathbb{E}[\nabla\ell(z, \theta_i)|\theta_i] = \nabla\mathcal{L}(\mathcal{D}, \theta_i)$.*

3. **Bounded variance**: *The variance of the stochastic gradient is bounded by a constant $\sigma$,*

$$\forall i : \mathbb{E}[\|\nabla\ell(z, \theta_i) - \nabla\mathcal{L}(\mathcal{D}, \theta_i)\|^2 |\theta_i] \leq \sigma^2 \,.$$

**Theorem 4.1.** *Under the above assumption, applying SGD with DropCompute (Algorithm 1), ensures*

$$\mathbb{E}\|\nabla\mathcal{L}(\mathcal{D}, \bar{\theta})\|^2 \leq \frac{2Lb_{\max}(\mathcal{L}(\mathcal{D}, \theta_1) - \mathcal{L}(\mathcal{D}, \theta^*))}{K} + \frac{2\sigma\sqrt{L(\mathcal{L}(\mathcal{D}, \theta_1) - \mathcal{L}(\mathcal{D}, \theta^*))}}{\sqrt{K}} \,, \quad (2)$$

*where $b_{\max}$ is the maximal total batch size, $K$ is the total number of samples that are used throughout the training, $\theta_1$ is the initial model, and $\bar{\theta}$ is a random sample of $\theta_i$ from the trajectory obtained by Algorithm 1, where $i$ is selected with probability proportional to the total batch size at iteration $i$. The expectation is with respect to the randomization introduced due to sampling from $\mathcal{D}$ throughout the optimization process and with respect to choosing $\bar{\theta}$.*

The bound in Theorem 4.1 is similar to existing fixed batch-size guarantees (Dekel et al., 2012). The second term in the bound does not degrade with the batch sizes, and behaves like $O(1/\sqrt{K})$, while the first term behaves like $O(b_{\max}/K)$. This implies that as long as $b_{\max} \leq O(\sqrt{K})$, the second term is dominant and we are in the regime of linear speedup. This shows we can attain a linear speedup despite using changing batch sizes, as long as the maximal batch size is bounded.

Similarly, in Theorem D.1 in appendix D.1 we show that the loss itself converges, in the convex case along with a proof. The proof of Theorem 4.1 is provided in appendix D.2. Lastly, we discuss the impact of the stochastic batch size on generalization in appendix D.4.

## 4.2 Iteration time in standard synchronous distributed training

We start with finding a closed-form expression for the cumulative distribution function (CDF) of the iteration time, denoted as $T$, defined as

$$T = \max(T_1, T_2, ..., T_N) \,,$$

---
[2]This is easily satisfied when all workers can access all data.

where $T_n$ represents the time taken by worker $n$ to compute its local batch, which follows some cumulative probability function $F_{T_n}(x) = \mathbb{P}(T_n < x)$, which we assume is independent for each worker. Let $F_T(x)$ represent the cumulative distribution function and $f_T(x)$ represent the probability density function of the maximum iteration time $T$. The relation between $F_T(x)$ and $F_{T_n}(x)$ is

$$F_T(x) = \mathbb{P}\left(\max\left(T_1, \ldots, T_N\right) \leq x\right) = \prod_{n=1}^{N} F_{T_n}(x).$$

Differentiating with respect to $x$ and applying the chain rule gives:

$$f_T(x) = \frac{dF_T}{dx}(x) = \sum_{n=1}^{N} f_{T_n}(x) \prod_{n' \neq n}^{N} F_{T_{n'}}(x).$$

In the special case where all of the workers' iteration time distributions are identically and independently distributed (i.i.d.), this reduces to the well-known formula:

$$f_T(x) = N \cdot f_{T_n}(x) \cdot F_{T_n}(x)^{N-1} \tag{3}$$

If the iteration time of each worker is distributed normally ($\sim \mathcal{N}(\mu, \sigma^2)$), the expected value of $T$ can be approximated as shown by Bailey et al. (2014):

$$\mathbb{E}(T) \approx \sigma \cdot \left((1-\gamma) \cdot \Phi^{-1}\left(1 - \frac{1}{N}\right) + \gamma \cdot \Phi^{-1}\left(1 - \frac{1}{e \cdot N}\right)\right) + \mu \tag{4}$$

where $\Phi$ is the CDF of the standard normal distribution, and $\gamma$ is the euler-mascheroni constant. Asymptotically the total iteration time is $\mathbb{E}[T] = \Theta(\sqrt{\log N})$. When the number of micro-batches $M \gg 1$, we can make a similar approximation under Central Limit Theorem (CLT) conditions. More details are in appendix C.2.

## 4.3 Iteration time and number of micro-batches with *DropCompute*

When using *DropCompute* with a constant threshold $\tau$, each worker is preempted at $\tilde{T}_n = \min\{\tau, T_n\}$ and joins the *AllReduce*. Therefore, the total iteration time with *DropCompute* is

$$\tilde{T} + T^c = \min\{\tau, T\} + T^c,$$

where $T^c$ is a serial latency present in each iteration, which includes the *AllReduce* step. This upper limit serves to clip extreme values of $T_n$, effectively constraining the range of potential outcomes for $\tilde{T}$. As a result, the compute time variability decreases, leading to a narrower distribution and enhanced compute efficiency. These effects are illustrated in Figure 2.

As a consequence of preempting each worker at $\tilde{T}_n$, the number of micro-batches computed in each step varies. Denote as $t_n^{(m)}$ the compute latency of a single micro-batch $m$ for worker $n$, and $T_n^{(m)} = \sum_{j=1}^{m} t_n^{(j)}$. We can define the average number of micro-batches computed by each worker before reaching threshold $\tau$ as

$$\tilde{M} = \frac{1}{N} \sum_{n=1}^{N} \sum_{m=1}^{M} \left\{ \begin{array}{ll} 1, & \text{if } T_n^{(m)} < \tau \\ 0, & \text{otherwise} \end{array} \right\}.$$

Under CLT conditions, the expected value for $\tilde{M}$ can be approximated in a closed form:

$$\mathbb{E}[\tilde{M}] \approx \sum_{m=1}^{M} \Phi\left(\frac{\tau - m \cdot \mu}{\sqrt{m \cdot \sigma^2}}\right) \tag{5}$$

where $\mu, \sigma^2$ are the mean and variance for a single micro-batch $t_n^{(m)}$ compute latency, and $\Phi$ is the CDF of the standard normal distribution. This approximation closely fits the real value of $\tilde{M}$ and can be used to analyze the expected gain from *DropCompute*. More details in appendix C.2.

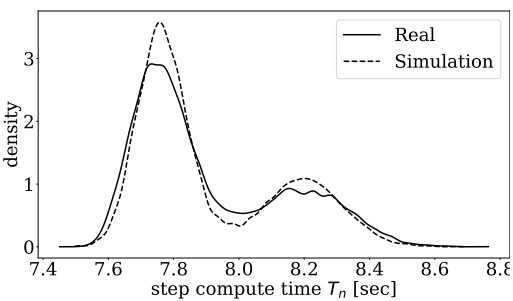
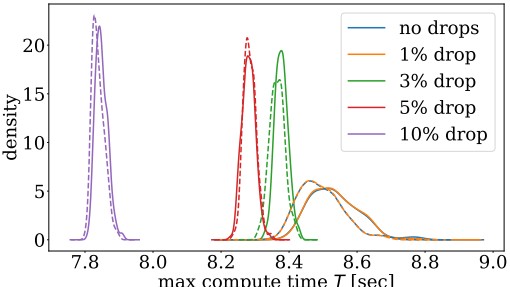

Figure 2: **Reduction in variance and mean iteration time using *DropCompute*.** Iteration time distribution of 200 workers using *DropCompute* on BERT 1.5B. (left) Step time $T_n$ distribution of all workers, without *DropCompute*. (right) Maximum iteration time $T$, across all workers, using *DropCompute*, with different drop rates. The dashed 'simulation' distribution is generated by drawing $T_n$ randomly from an independent normal distribution separately for each worker, using the empiric mean and variance of that worker.

### 4.4   Choosing the threshold

The throughput of the system can be seen as the number of micro-batches computed per second. For $N$ workers, this can be written as $NM/(T + T^c)$. To evaluate the effectivness of *DropCompute*, we consider the difference in throughput between the baseline and when using *DropCompute*. Doing so, we can define the effective speedup for $\tau$ as:

$$S_{\text{eff}}(\tau) = \frac{\text{DropCompute Throughput}}{\text{Baseline Throughput}} = \frac{N\tilde{M}/(\min\{\tau, T\} + T^c)}{NM/(T + T^c)} = \frac{\tilde{M}(T + T^c)}{M(\min\{\tau, T\} + T^c)} \quad (6)$$

Given the statistical characteristics of the training setting, it is possible to estimate analytically the expected value of the effective speedup $\mathbb{E}[S_{\text{eff}}(\tau)]$ by using Equations 5 and 4. Moreover, when plugging in the asymptotic form of $\mathbb{E}[T]$, we find the expected speedup increases to infinity with $N$

$$\mathbb{E}[T] = \Theta(\sqrt{\log N}) \quad \Rightarrow \quad \mathbb{E}[S_{\text{eff}}(\tau)](N) \xrightarrow[N \to \infty]{} \infty$$

As shown in figure 3b, Equation 4 is less accurate when samples deviate from a normal distribution. To find the optimal compute threshold, we synchronize the empirical distribution of micro-batch

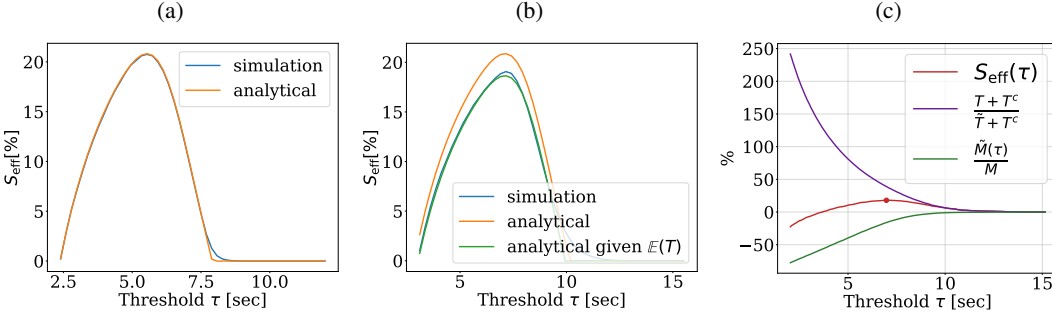

Figure 3: **Statistical characteristics of the micro-batch computation latency $t_n^{(m)}$ can provide a reliable estimate of the effective speedup $S_{\text{eff}}$.** The graphs depict $S_{\text{eff}}(\tau)$ based on Equation 6. Calculations rely on samples of $t_n^{(m)}$ to calculate $\tilde{M}$ and $T$, before plugging them into Equation 6. In the 'simulation' curves, we directly use the samples to calculate $\tilde{M}$ and $T$. For the 'analytical' curves, only the mean and variance of $t_n^{(m)}$ are used to approximate $\tilde{M}$ and $T$ using Equations 5 and 4, respectively. For 'analytical given $\mathbb{E}[T]$', $\tilde{M}$ is approximated using Equations 5, but $\mathbb{E}[T]$ is calculated directly from the samples. More details in appendix C.2. Panels: (a) The $t_n^{(m)}$ samples follow a normal distribution. (b) samples are taken from BERT1.5B pre-training with simulated delay as described in Section 5.2. (c) **The optimal compute threshold can be found automatically.** The effective speedup, micro-batch completion rate, and step speedup as a function of $\tau$ in a simulated delay environment.

compute latency between all workers after a few iterations. Given this distribution, we find $T$ and $\tilde{M}$, and search in a decentralized way for the threshold $\tau^*$ that maximizes the effective speedup $S_{\text{eff}}(\tau)$

defined in Equation 6. Overall, the cost of synchronizing the empirical distribution and finding $\tau^*$ is negligible, compared to a full training session, because it happens only once in a training session. Lowering the threshold leads to reduced compute time but higher compute drop rates. Figure 3c highlights this trade-off and the optimal $\tau^*$ is marked.

### 4.5 Compensating for dropped samples

The effective speedup metric $S_{\text{eff}}$, accounts for dropped samples by treating them as a source of slowdown in direct proportion to the drop rate. This consideration enables us to execute additional computations to achieve the theoretical speedup. The extent of extra time spent on redundant calculations can be as much as $R = (M/\tilde{M} - 1)$ times the computational effort required when not applying *DropCompute*.

For instance, when 10% of the samples are dropped, we can expect to perform approximately 11% more calculations. Achieving this can be approached in several ways. One straightforward compensation method for LLM training involves adding an extra $R \cdot I_{\text{base}}$ steps to the training process, where $I_{\text{base}}$ represents the number of training steps conducted without using *DropCompute*. In practice, achieving the original accuracy often requires even fewer additional steps, as illustrated in Figure 5, resulting in an even higher effective speedup.

Another method of compensating for the dropped samples is to increase the maximal batch size. When increasing the batch by $R$ and dropping in average $1 - \tilde{M}/M$, we keep the average batch size the same as without using *DropCompute*, hence compensating for the lost samples. A third method, orthogonal to the first two, is resampling dropped samples before starting a new epoch to diversify the overall samples seen by the model. These approaches are rigorously tested and compared in Table 1b

## 5 Experiments

To be useful, *DropCompute* must possess two properties. First, it should not compromise the accuracy of the trained model. This property is put to test in section 5.1 where we fully train BERT-Large and ResNet-50 (Devlin et al., 2018; He et al., 2015), each on a different task, with different drop rates to compare accuracy. Second, *DropCompute* should maintain a high level of runtime performance, especially when compute variance or straggling workers exist and vanilla synchronous training time deteriorates. Section 5.2 tests runtime performance of *DropCompute* by training a 1.5 billion parameter language model, BERT1.5B (Devlin et al., 2018) with additive noise to the compute time of each worker.

**Experimental setup.** The analysis of all BERT models is performed on the same dataset as Devlin et al. (2018), which is a concatenation of Wikipedia and BooksCorpus with 2.5B and 800M words respectively. The finetuning of the pretrained models is performed on SQuAD-v1 (Rajpurkar et al., 2016). We verify the generality of *DropCompute* by additional evaluation of a ResNet-50 model for image classification on ImageNet (Deng et al., 2009). The experiments depicted in section 5.2 and section 5.1 are executed on Habana Gaudi-1 and Gaudi-2 accelerators, respectively, with high performance network (Habana, 2020).

### 5.1 Generalization performance

The sole difference in the optimization when *DropCompute* is applied is that the batch size is not deterministic, but stochastic, as explained in section 3.2. To complement theorem 4.1, we examine the generalization performance achieved with a stochastic batch size on two popular tasks.

**Image classification.** To evaluate the generality of stochastic batch size and *DropCompute* in particular, we evaluate the Top-1 accuracy of a ResNet-50 model on the Imagenet dataset using our method. Since it is not common to use gradient accumulation in large scale training of this task, we simulate the drops such that each worker randomly drops its local batch, so the total batch size is stochastic. This simulated environment enables us to examine the extent of drop rate we can use without compromising accuracy. Figure 10 in appendix B.2.2 shows that up to $10\%$ drop rate, which is more than what *DropCompute* operates on, there is a negligible deterioration in accuracy.

**Large language model.** Training LLMs is resource intensive, typically using large batch sizes, which makes *DropCompute* appealing. We evaluate *DropCompute* method on this task by fully pretraining

| (a) Varying drop rate, no compensation | | (b) 10% drop rate, with compensation | |
| --- | --- | --- | --- |
| % Drop rate | F1 score on dev set | Compensation method | F1 score on dev set |
| 0% | $91.32 \pm 0.15$ | None | $91.19 \pm 0.02$ |
| 2.5-3% | $91.34 \pm 0.04$ | 11% extra steps | $91.40 \pm 0.08$ |
| 5.5-6% | $91.44 \pm 0.02$ | 11% increased batch size | $91.38 \pm 0.08$ |
| 10-11% | $91.19 \pm 0.02$ | re-computation | $91.19 \pm 0.11$ |

Table 1: **Maintaining the accuracy of BERT-Large pretraining.** Fine-tuning results on SqUAD v1.1, where the F1 score is obtained by the pretrained model. **(a)** The effect of different drop rates during pretraining on the final accuracy, without compensating for the dropped samples. **(b)** When 10% drop rate is used during pretraining, with different methods of compensating for the dropped samples.

BERT-Large model several times, each with a different drop rate. We follow the optimization regime described in You et al. (2019) with a batch size of 64K for phase-1 and 32K for phase-2 (more details are provided in appendix B.2). Each of the pretrained models is fine-tuned on the SQuAD task 3 times with different initializations. Fine-tuning is performed without drops, as it is not a large scale resource consuming task. Table 1a shows the average accuracy ($\pm$ standard deviation) obtained for each drop rate. As shown, *DropCompute* at drop rates of up to $10\%$ have negligible accuracy difference. Higher values measured up to $20\%$ of dropped gradients provide acceleration with a small yet discernible hit on accuracy. We note that these results are for a fixed budget of steps. In the presence of compute variance, the effective speedup indicates that additional steps can be executed while still maintaining competitive runtime performance. This notion is demonstrated in section 5.2.

## 5.2 Runtime performance

The main purpose of our proposed method is to maintain runtime performance when compute variance is present. We examine this by measuring the speedup of *DropCompute* over standard synchronous training in several settings. First, we measure the potential speedup for different drop rates and training settings by post analysis of synchronous training without drops. In addition, we introduce compute variance by training with additive noise, and measure actual speedups using *DropCompute*. The experiments in this section are performed on BERT1.5B. Details are provided in appendix B.1.

**Training with different number of workers and micro-batches.** We evaluate the potential speedup of *DropCompute* on several training settings with natural heterogeneity and no drops. For each setting, we post analyze what would have been the speedup for different drop rates. As can be seen in Figure 4, *DropCompute* exhibits increasing benefits with a growing number of workers and compute requirements. However, there are diminishing returns in terms of speedup with more accumulations. This could possibly be explained by the amortization time of a large number of micro-batches.

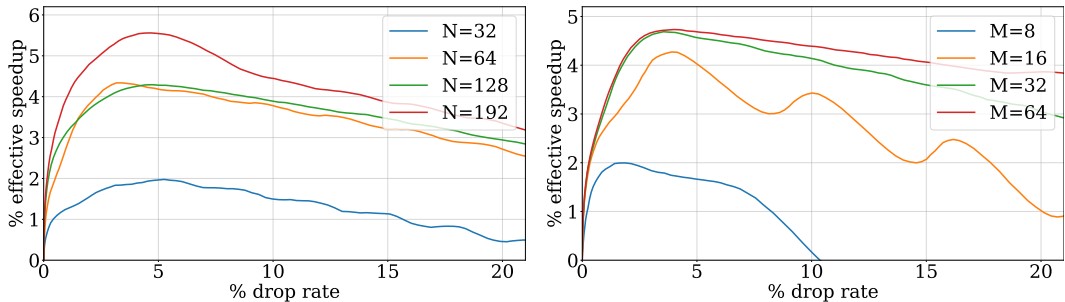

Figure 4: *DropCompute* **exhibits increasing benefit on a large scale.** Effective speedup versus drop rate with (left) 32 accumulations and varying workers, and (right) 112 workers and varying number of accumulations.

**Simulated delay environment.** Although *DropCompute* may have value when the workers' compute latency variance is low, its significance becomes crucial when the workers' compute latency exhibits high variability. To evaluate our method, we introduce a delay environment where random latency is added to each micro-batch computation. This additive noise follows a bounded log-normal distribution. Detailed information and motivation regarding the additive noise are in appendix B.1. The experiments are executed with 12 gradient accumulations and a local batch size of 192. In Figure 1, the negative impact of compute variance on scalability is demonstrated and mitigated

using *DropCompute*. The results in Figure 1 also correspond to section 4 and Equation 11, where a theoretical extrapolation follows the same trend line. When utilizing *DropCompute* in this setup, achieving the same training loss as the baseline might requires additional training steps, however, it leads to a notable reduction in overall training time. Figure 5 demonstrates it in a training session with 64 workers, where approximately 3% more steps is needed to reach the same loss, in 13% less time.

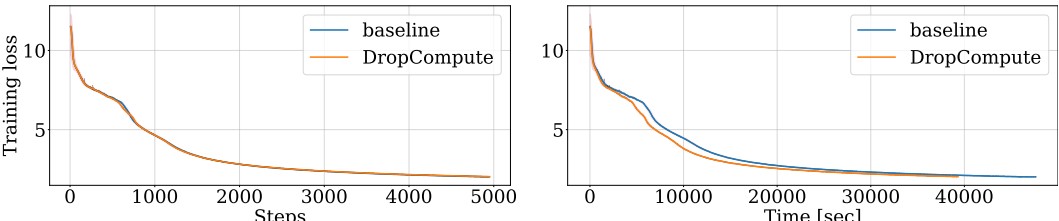

Figure 5: *DropCompute* **improves training time for workers with compute variance.** Train loss curve of BERT1.5B pretraining, in a simulated delay environment. (left) Horizontal axis in training steps, and (right) horizontal axis in training time.

## 6 Discussion

**Summary.** Efficient scalable systems are a key component to enable the continued development of deep learning models. To this day, state-of-the-art models rely on synchronous distributed optimization. The challenge to maintain synchronous training as an efficient solution grows larger with the quickly growing model sizes and data. Therefore, improving the robustness and scalability of distributed synchronous training is an important endeavor. This paper tackles the challenge of maintaining synchronous training scalable in the face of compute variance. We propose *DropCompute* to improve the robustness of synchronous training. Workers drop their remaining compute when they reach a compute threshold, determined by exchanging and analyzing the compute latency distribution. We find that for a small percentage of dropped data, a much larger percentage of time can be saved, depending on the compute latency distribution of the workers. In addition, we provide theoretical convergence guarantees and runtime predictions. We further discuss the motivation behind *DropCompute* and how it effectively solves the problem in appendix A.

**Limitations.** While *DropCompute* is simple and straightforward, it deals with system efficiency, and as such, the user-level implementation provided is not optimal. Mainly, the provided implementation is limited by using many gradient accumulations and integrating compute timeout in between them. However, we believe that this is not a major concern since having multiple gradient accumulations is a common practice in training LLM on a large scale and is used in state-of-the-art training configurations (Smith et al., 2022; Nvidia, 2023). In addition, *DropCompute* addresses variance that originates from the compute stage of the training iteration and does not solve the potential issue of network variance during the all-reduce stage.

**Future directions.** *DropCompute* is described and analyzed in this paper as a method built on top of synchronous training. However, this method can be integrated with other possibly asynchronous methods such as periodic synchronization. In appendix B.3, we implement *DropCompute* on top of Local-SGD (Lin et al., 2020) and show that *DropCompute* can also improve the robustness of Local-SGD to stragglers. A different extension for *DropCompute* is to apply it during the model backward calculation and save the partial gradients that were already calculated. This would generalize *DropCompute* for workloads that do not utilize gradient accumulations. However, it will require further study as it differs from the stochastic batch-size setting where the entire data sample is either saved or dropped.

## Acknowledgments

We thank Itay Hubara for technical advising and valuable comments on the manuscript. The research of DS was Funded by the European Union (ERC, A-B-C-Deep, 101039436). Views and opinions expressed are however those of the author only and do not necessarily reflect those of the European Union or the European Research Council Executive Agency (ERCEA). Neither the European Union nor the granting authority can be held responsible for them. DS also acknowledges the support of Schmidt Career Advancement Chair in AI.

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
