# Appendix

## A  Further discussion

In this section, we will further elaborate on the motivation for using *DropCompute* and how it mitigates existing problems in large-scale training.

### A.1  Motivation

The primary objective of *DropCompute* lies in the mitigation of compute latency variance among workers. This raises the question of the significance of compute variance in the context of our research. Compute variance can arise from various sources, including but not limited to faulty hardware, clock throttling, host preemption/overhead, inefficient load balancing, connectivity issues (particularly in model parallel settings), and more. Inefficient load balancing is especially in particular, when dealing with with dynamic sentence/image sizes (Tan & Le, 2021; Dehghani et al., 2023; Raffel et al., 2020b) because it requires special treatment for each model and data set, often at the expense of performing redundant work. Addressing these issues typically requires intricate engineering efforts:

- Regular testing and replacement of faulty hardware.
- Mitigation of host overhead through the implementation of latency-hiding techniques and script optimization.
- Management of inefficient load balancing on a per-workload basis, employing strategies such as sample padding and packing.

However, it is essential to recognize that each of these issues represents a potential single point of failure. The triggering of any one of them can result in a substantial performance degradation within large-scale systems. Some of our early experiments exhibited naturally such sub-optimal behavior where we clearly see a large variance in compute latency between iterations and workers (as shown in figure 6). As our goal is to improve robustness (i.e. performance for outlier cases), these cases are important. Moreover, after the compute variance was reduced by HW and SW optimizations, we were still left with some compute variance (as shown in figure 2).

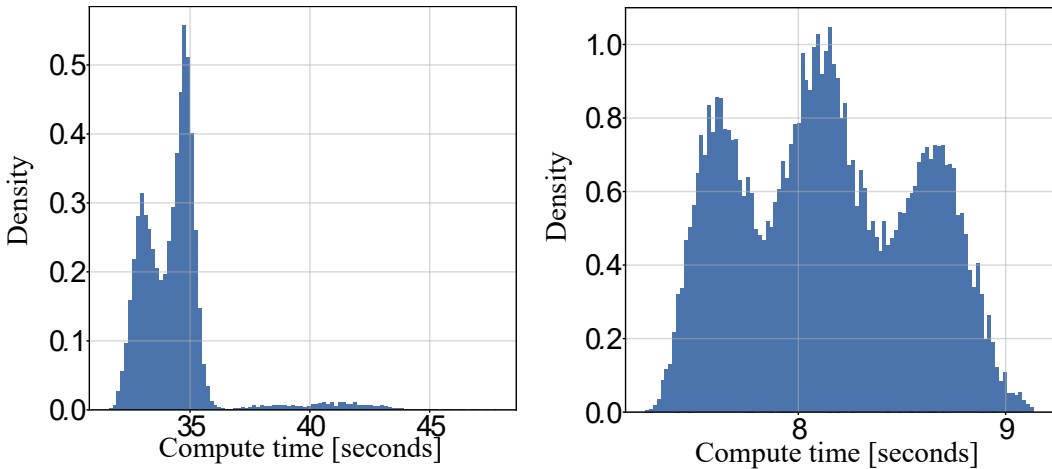

Figure 6: **Single iteration latency histogram in a sub-optimal system.** These histogram were recorded in BERT1.5B training, before optimizing our system. (left) Training with 162 workers and 64 gradient accumulations. When applying *DropCompute* in this setting, we achieved $\sim 18\%$ performance boost. (right) Training with 190 workers and 16 gradient accumulations.

These examples lead us to the conclusion that, in practice, workers do not finish computation at the same time, and this can have a significant impact on the training speed. Moreover, the effect of stragglers and compute variance on the training speed is expected to get worse as the distributed scale increases. This is due to the maximal worker distribution relation stated in equation 3. When

modeling the additive latency as normally distributed, the average maximal worker latency, $\mathbb{E}[T]$, increases with the number of workers $N$ as $\mathbb{E}[T] = \Theta(\sqrt{\log N})$ as shown in appendix C.2.

## A.2 Effectiveness

Mitigating slowdowns resulting from compute variance can be achieved easily buy using *DropCompute*. For instance, in a sub-optimal system with stragglers, as illustrated in Figure 6 (left), we were able to recover approximately 18% of the runtime performance. The contribution of *DropCompute* can be even more significant in different systems with varying noise distributions, as demonstrated in Appendix C.3.

Furthermore, even when assuming a normal distribution of noise, theoretical analysis indicates a substantial speedup as the number of workers $N$ increases: $S_{\text{eff}}(N) \xrightarrow[N \to \infty]{} \infty$. On the tested system, after reducing compute variance through hardware and software optimizations, we achieved a 5% performance boost with 196 workers (see Figure 2). Notably, this speedup continues to increase as the scale of the system grows. These examples underscore how *DropCompute* enhances the robustness of large-scale training, effectively recovering lost performance attributed to stochastic performance outliers.

Lastly, it's important to emphasize that even a modest improvement in large-scale training performance can yield significant cost savings. For example, assuming a cost of \$10–32/hour/8xA100 (according to AWS pricing), saving 5% of the training time for a 176B model, such as (Scao et al., 2022), would result in savings ranging from \$67,686 to \$216,598. For longer-trained models, like (Touvron et al., 2023), the savings could reach \$107.50 to \$344,064."

# B Experiments

## B.1 Runtime performance experiments

In this section, we provide details for the experiments of section 5.2.

**Experiment details.** As mentioned in the paper in section 5.2, we pre-train BERT1.5B following Habana (2023). The experiments in this section use up to 200 Gaudi accelerators with high bandwidth inter-connectivity. The training is done with a maximum input sequence length of 128 and 80 maximum predictions per sequence. The training regime consists of a local batch size of 196, 12 gradient accumulations, LANS optimizer (Zheng et al., 2020), and a learning rate of 0.0015. Due to the large capacity of the model, we used ZeRO optimizer stage 1 to fit the model in memory (Rajbhandari et al., 2020).

**Simulated delay.** Many frameworks use padding to allow for constant input length which improves hardware efficiency (Kosec et al., 2021). However, some learning tasks inherently involve dynamic shapes, such as translation (Ott et al., 2018) and multi-task sequences (Raffel et al., 2020a). These use cases motivate us to explore scenarios of dynamic length via simulation. To demonstrate the value of *DropCompute* in dealing with compute variance we added to each micro-batch compute time an additional random waiting time. The additive noise is based on a Log-normal distribution since it is typical for user post lengths in internet discussions (Sobkowicz et al., 2013), which are used as training data in recent language models (Radford et al., 2019). To make this setting more realistic, we scale down and bound the noise so that each accumulation takes $\times 1.5$ longer on average, and, in extreme cases, can take up to 6 times longer. This allows us to simulate stragglers and high compute variance while keeping a conservative limit on iteration time. Thus, the additive noise takes the form of

$$\epsilon = \min\left(\frac{1}{\alpha} Z, \beta\right), \qquad Z \sim \text{LogNormal}(4, 1).$$

This noise was added to each accumulation

$$t_n^{(m)} \leftarrow t_n^{(m)} + \mu \cdot \epsilon,$$

where $\mu$ is the mean value for $t_n^{(m)}$, $\alpha = 2 \exp(4.5)$ and $\beta = 5.5$ are the scaling and bounding constants, and the log-normal parameters (4,1) fit user post lengths, as seen in Sobkowicz et al. (2013). As illustrated in Figure 7, the noise distribution leads to each micro-batch latency increased by up

to $6\mu$, while the majority of accumulations have low latency. Further analysis on the effect of noise properties is discussed in C.3.

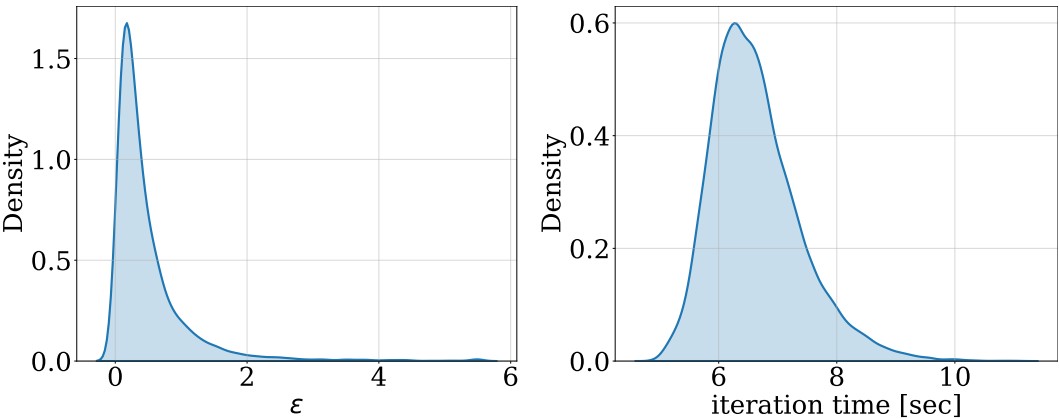

Figure 7: **The latency distribution in a simulated delay environment.** (left) The distribution of the additive noise $\epsilon$, added to each accumulation. (right) The distribution for iteration time $T_n$, with 12 accumulations, each with added noise, in BERT1.5B training.

## B.2 Generalization experiments

In this section, we provide details for the experiments of section 5.1.

### B.2.1 Large language models

Here we provide more details about how the LLM experiment was executed as well as additional graphs related to the LLM experiment described in section 5.1.

**Experiment details.** As mentioned in the paper, in section5.1 we follow You et al. (2019) optimization regime with LAMB optimizer. Specifically, for phase-1 where the sequence length is 128 tokens per sample, we use a batch size of 64K, the learning rate is 0.006, the warmup ratio is 0.2843, and the steps number is 7038. For phase-2 where the sequence length is 512, we use a batch size of 32K, the learning rate is 0.004, the warmup ratio is 0.128 and the steps number is 1563. The experiments were executed on 64 workers.

**Batch size distribution.** As explained in section 5.1 we fully pretrain a BERT-Large model with *DropCompute* several times, each with a different drop rate. Figure 8 shows the empirical batch distribution of each of the drop rates in phase-1.

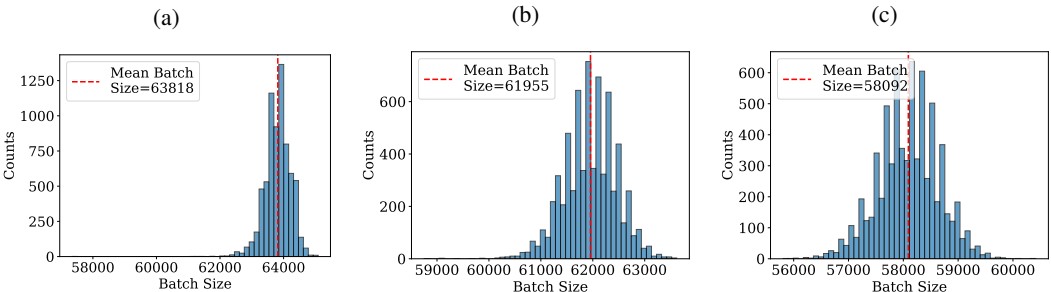

Figure 8: **Batch size distribution.** BERT-Large phase-1 pretraining batch size distribution when using *Drop-Compute* and drop rate of (a) 2.5% , (b) 5.5%, and (c) 11.5%

**Convergence loss.** In addition to the results depicted in Table 1a, we show the convergence of the training loss with the different drop rates in Figure 9.

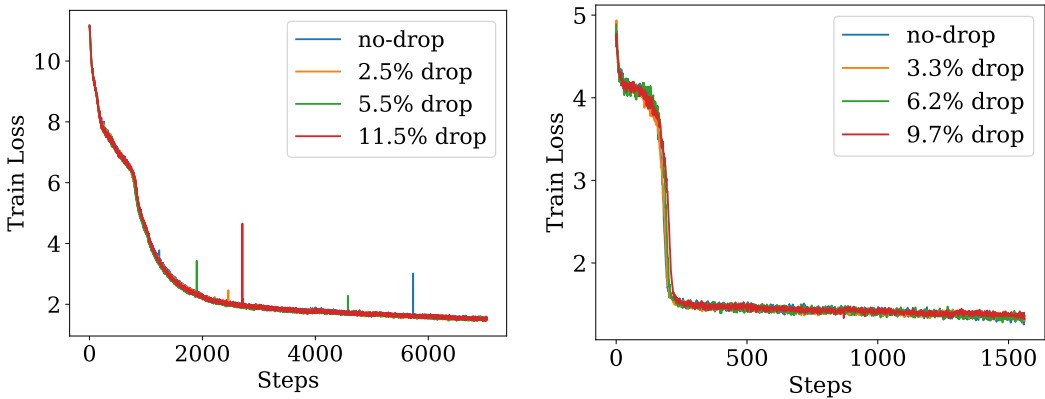

Figure 9: **Train loss convergence.** BERT-Large phase-1 (left) and phase-2 (right) pretraining train loss for different drop rates.

### B.2.2 Image classification

This section provides the details of the image classification experiment described in section 5.1 as well as Figure 10 which is referenced from the paper.

**Experiment details.** To simulate *DropCompute*, at each training step, the gradients of each worker are set to zero with a probability of $P_{\text{drop}}$. We repeat each training process 3 times with different initializations. To examine the generalization of *DropCompute* over different optimizers, we implement our method on two popular training regimes of ResNet50. First, we follow the optimization regime described in Goyal et al. (2017) that uses SGD with 32 workers and a global batch size of 4096. Second, we follow Mattson et al. (2019) that uses LARS (You et al., 2017) with 8 workers and a global batch size of 2048.

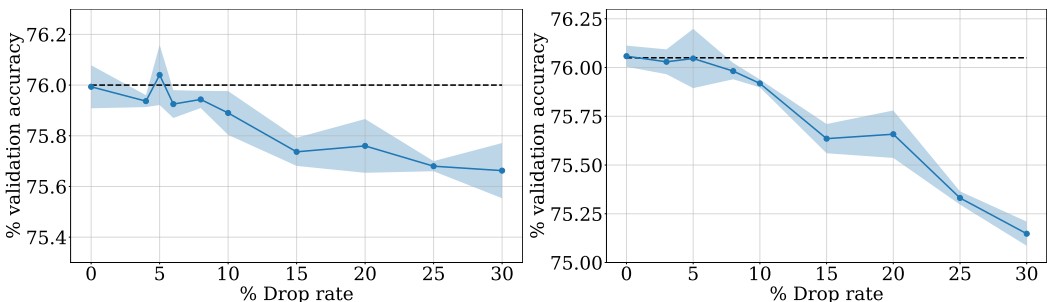

Figure 10: **Generalization over varying drop rates.** Top-1 validation accuracy of ResNet50 trained on ImageNet with varying simulated drop rates. The dashed line is the baseline accuracy without drops. The solid line is the average over 3 runs, and the blue area is the standard deviation. (left) Training regime with SGD (Goyal et al., 2017). (right) Training regime with LARS (Mattson et al., 2019). Up to a 10% drop rate, there is a negligible accuracy deterioration.

**Learning rate correction.** Previous works showed that the learning rate should be scaled with respect to the batch size (Hoffer et al., 2017; Goyal et al., 2017). With a stochastic batch size and specifically *DropCompute*, it is possible that a learning rate correction should be considered to maintain accuracy with the same number of steps. We examine such corrections when training with stochastic batch size. First, we decrease the learning rate by a constant factor, equal to the average drop rate. Specifically, for an average drop rate $P_{\text{drop}} \in [0, 1]$ we multiply the learning rate by $(1 - P_{\text{drop}})$. A different correction we consider is a stochastic correction, such that in each step we divide the gradients by the computed batch size, instead of the original batch size. This result in a different normalization in each step depending on the actual dropped samples. We note that for the latter, the workers have to synchronize the computed batch of each worker at each step. This is generally can be done during the *AllReduce*, with negligible overhead. We repeat the training of ResNet50 on ImageNet as described in Goyal et al. (2017) to evaluate the generalization without correction and with the two suggested

corrections. We use 128 workers, batch size 8192, and use ghost batch norm (GBN) (Hoffer et al., 2017) to match batch normalization of 32 samples and restore the results in Goyal et al. (2017). As can be seen in Figure 11, for low drop rates, there is no superior correction method, and no correction generally achieves the same generalization. Yet, it is possible that a learning rate correction could potentially improve generalization on a different task or with a different optimizer.

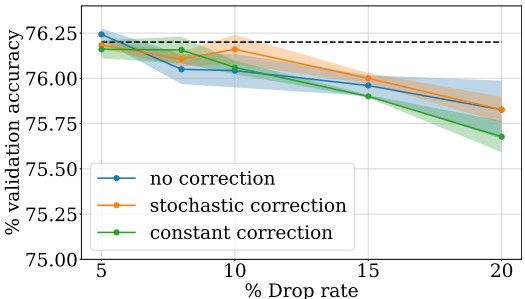

Figure 11: **Learning rate correction is not necessary for low drop rates.** Top-1 validation accuracy of ResNet50 trained on ImageNet with varying simulated drop rates, using the learning-rate correction methods described in B.2.2. The dashed line is the baseline accuracy without drops. Each solid line is the average over 3 runs, and the area around it is the standard deviation. Up to a 10% drop rate, there is a negligible accuracy deterioration regardless of the correction applied.

## B.3 Local-SGD

Periodic synchronization methods, such as Local-SGD, provide better scalability properties than synchronous methods. By exchanging parameters less frequently, communication overhead is mitigated. For compute variance and straggling workers in particular, the robustness of these methods greatly depends on the distribution of the compute time between workers. For example, when straggling workers appear randomly with homogeneous distribution, Local-SGD can mitigate the straggling workers slowdowns to some extent; this is because of the amortization effect in synchronizing periodically once every several steps. On the other hand, if straggling workers appear from a small set of workers such as a single server, a realistic scenario, Local-SGD acts more closely to synchronous training as the worst-case scenario is when a single worker always straggling behind. *DropCompute* can be easily integrated with Local-SGD by leveraging periodic synchronization instead of gradient accumulations. We implement *DropCompute* on top of Local-SGD by comparing the compute time with a threshold at each local step. We show that when stragglers are apparent, *DropCompute* can improve the robustness of Local-SGD. We randomly slow down workers to simulate stragglers in two scenarios as described in Figure 12. The experiment setting is 32 workers training on ResNet50 and ImageNet. At each local step, each worker is selected to be straggler with a $4\%$ chance. This way, there is at least 1 straggler for each local step on average. We measure relative speedup compared to synchronous training in terms of step time, both for Local-SGD and with *DropCompute* on top of Local-SGD. As can be seen, with *DropCompute* (set to $6.2\%$ drop rate in this experiment) we improve the robustness of Local-SGD.

## C Analyzing the effective speedup using *DropCompute*

In this section, we provide more details on the process of choosing the threshold $\tau^*$ that will maximize the effective speedup. We begin by giving technical details on the process used during training, given samples drawn from the empirical distribution of the compute latency. Next, we continue to explore and establish the analytic connection between the latency statistics and the effective speedup.

### C.1 Automatic selection of the drop threshold

In Algorithm 2 below we present the algorithm used for automatic selection of the drop threshold $\tau$. In this algorithm, $t_{i,n}^{(m)}$ is the time it takes worker $n$ to process micro-batch $m$ at step $i$ and $T_i^c$ is the time spent on communication for at step $i$. This data is measured by each worker, and then synchronized between all workers after $I$ iterations. After the synchronization step each worker will

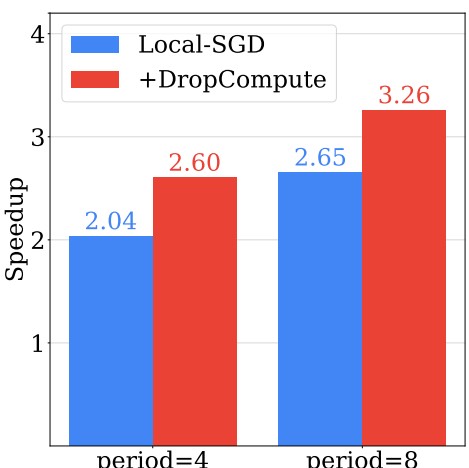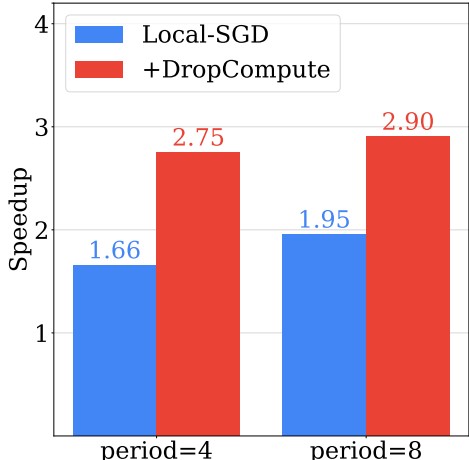

Figure 12: ***DropCompute* can be integrated with Local-SGD to improve robustness.** Speedup of each method compared to synchronous training in a straggling workers environment. In each local step workers are randomly drawn to be stragglers. If selected as a straggler, the worker waits 1 second before continuing. The synchronization period of Local-SGD is denoted in the x-axis. (left) Uniform stragglers. (right) Single server stragglers.

have the same drop threshold $\tau$, which depends on both his own speed and the compute latency of the other workers. Since $T_i^c$ is used in the normalization of the effective speedup, the chosen threshold takes into account both compute and communication time.

---

**Algorithm 2** Automatic choice for the optimal threshold $\tau$

---

**Input**:
number of workers $N$
number of iterations $I$
number of micro-batches per step $M$
micro-batch time samples $\{\{t\}_{i,n}^{(m)}\}^{i \in [1:I], n \in [1:N], m \in [1:M]}$
communication time for each iteration $T_i^c$
potential thresholds $[\tau_0, \tau_1, ...]$
**for** $\tau \in [\tau_0, \tau_1, ...]$ **do**
    **for** $i = 1$ to $I$ **do**
        Initialize completed micro-batch count: $\tilde{M}_i(\tau) = 0$
        Initialize compute step latency for all workers: $T_i = 0$
        **for** $n = 1$ to $N$ **do**
            Initialize single worker step compute latency: $T_{i,n} = 0$
            **for** $m = 1$ to $M$ **do**
                $T_{i,n} \leftarrow T_{i,n} + t_{i,n}^{(m)}$
                $\tilde{M}_i(\tau) \leftarrow \tilde{M}_i(\tau) + \frac{1}{N} \cdot \left\{ \begin{array}{ll} 1, & \text{if } T_{i,n} < \tau \\ 0, & \text{otherwise} \end{array} \right\}$
        $T_i \leftarrow \max(T_i, T_{i,n})$             ▷ compute time of the slowest worker at step (i)
        $S_i(\tau) = \frac{T_i + T_i^c}{\min(\tau, T_i) + T_i^c} \cdot \frac{\tilde{M}_i(\tau)}{M}$         ▷ Effective speedup for step (i)
    $S_{\text{eff}}(\tau) = \frac{1}{I} \sum_{i=1}^{I} S_i(\tau)$           ▷ Mean speedup for threshold ($\tau$)
$\tau^* \leftarrow \text{argmax}_\tau \left( S_{\text{eff}}(\tau) \right)$

---

## C.2 DropCompute speedup analytic analysis

In this section we further explore the relation between the compute latency distribution and the effective speedup. We will derive a closed-form representation of the effective speedup by making certain assumptions. First we assume that

**Assumption C.1.** $t_n^{(m)}$ is i.i.d with finite mean $\mu$ and finite variance $\sigma^2$.

Note that in the assumption above, for simplicity of presentation we assumed all workers as identical and so $\mu$ and $\sigma^2$ are identical. However, it is possible to derive similar properties with nonidentical workers, each with their own $\mu_n$, $\sigma_n$. Next, denote the time for completion of micro-batch $m$ as $T_n^{(m)} = \sum_{j=1}^{m} t_n^{(j)}$. Then, we assume

**Assumption C.2.** $T_n^{(m)}$ is Gaussian $\sim \mathcal{N}(m\mu, m\sigma^2)$ for $m > \sqrt{M}$.

This assumption holds in the limit $M \to \infty$ given Assumption C.1, from the Central Limit Theorem (CLT). Lastly, denoting $\tau$ as the threshold used, we assume

**Assumption C.3.** $\tau > \frac{M\mu}{2}$.

This bound can be considered as the minimum threshold allowed, in order for *DropCompute* to be effective. Taking a lower threshold will result in unacceptable high drop rate.

Using these assumptions we first derive analytical expression for the iteration time $\mathbb{E}[T]$ and the mean completed number of gradient accumulations $\mathbb{E}[\tilde{M}]$ with *DropCompute*. Then, we combine these expressions to obtain an expression for the mean effective speed $\mathbb{E}[S_{\text{eff}}]$.

Figure 3b shows an example of how close is the derived expression of $\mathbb{E}[S_{\text{eff}}]$ to the value calculated by using the algorithm described in section C.1. The 'analytical' curve is slightly off due to the inaccuracy of the Gaussian Assumption C.2 in calculating $\mathbb{E}[T]$, as we discuss below. We therefore added another curve 'analytical given $\mathbb{E}(T)$', which uses same the derived expression for $\mathbb{E}[S_{\text{eff}}]$ but replacing value of $\mathbb{E}[T]$, with the empiric mean: $\overline{T} = \frac{1}{I} \sum_{i=1}^{I} \max_n \left\{ T_{n,i}^{(M)} \right\}$ where: $i \in [1 : I]$ are the iterations measured.

**Iteration time.** as written in section 4.2, the iteration time for all workers is

$$T = \max_n \left( T_n^{(M)} \right) = T^c + \max_n \left( \sum_{m=1}^{M} t_n^{(m)} \right).$$

When $T_n^{(M)} \sim \mathcal{N}(M\mu, M\sigma^2)$. the expected value of $T$ can be approximated as (Bailey et al., 2014):

$$\mathbb{E}[T] \approx \sqrt{M\sigma^2} \cdot \left( (1-\gamma) \cdot \Phi^{-1} \left( 1 - \frac{1}{N} \right) + \gamma \cdot \Phi^{-1} \left( 1 - \frac{1}{e \cdot N} \right) \right) + M\mu + T^c. \quad (7)$$

We can derive the asymptotic behavior of Eq. 7 by:

$$\Phi(x) = \frac{1}{2} + \frac{1}{2}\text{erf}\frac{x}{\sqrt{2}} \sim 1 - \frac{1}{x\sqrt{2\pi}}e^{-x^2/2}$$

$$\Downarrow$$

$$\log(1 - \Phi(x)) \sim -\frac{x^2}{2} - \log(x\sqrt{2\pi}) \sim -\frac{x^2}{2}, \quad x \to \infty$$

$$\Downarrow$$

$$\Phi^{-1}(1 - y) \sim \sqrt{-2\log y}, \quad y \to 0^+$$

When plugging Equation 7 into this asymptotic approximation we are left with $\mathbb{E}[T] = \Theta(\sqrt{\log N})$.

It is worth noting that the distribution of $T$ is mostly affected by the tail of distribution of $T_n = T_n^{(M)}$ (as a consequence of Equation 3). Therefore, in practice the Gaussian Assumption C.2, and therefore Equation 7, can be inaccurate, especially for small $M$ and large $N$. It is therefore more accurate to use the real value of $\mathbb{E}[T]$, measured without *DropCompute*, to estimate the potential effective speedup. An example for the inaccuracy of this approximation can be seen in Figure 3b, when $T_n^{(M)}$ does not follow a normal distribution.

**Completed micro-batches.** The average number of micro-batch computed by a single worker $n$ when using *DropCompute* with threshold $\tau$, is:

$$\tilde{M}(\tau) = \frac{1}{N} \sum_{n=1}^{N} \sum_{m=1}^{M} \left\{ \begin{array}{ll} 1, & \text{if } T_n^{(m)} < \tau \\ 0, & \text{otherwise} \end{array} \right\}.$$

Its expected value can be written as:

$$\mathbb{E}[\tilde{M}(\tau)] = \sum_{m=1}^{M} P\left(T_n^{(m)} < \tau\right).$$

In order to use assumption C.2, we can split $\tilde{M}$ into 2 sums and derive a closed-form formula for the expected value:

$$\tilde{M}(\tau) = \sum_{m=1}^{\lfloor\sqrt{M}\rfloor} P\left(T_n^{(m)} < \tau\right) + \sum_{m=\lceil\sqrt{M}\rceil}^{M} P\left(T_n^{(m)} < \tau\right). \tag{8}$$

For the right term we can use assumption C.2 so that $P(T_n^{(m)} < \tau) = \Phi\left(\frac{\tau - m\cdot\mu}{\sqrt{m\cdot\sigma^2}}\right)$. For the left term, when $m < \sqrt{M}$ we use Markov inequality and assumption C.3 to show that

$$0 \leq P(T_n^{(m)} > \tau) \leq \frac{\mathbb{E}[T_n^{(m)}]}{\tau} = \frac{m\mu}{\tau} \leq \frac{2m}{M}.$$

In other words, when using *DropCompute* with low drop rates, $P(T_n^{(m)} < \tau)$ is very high for $m < \sqrt{M}$. The Gaussian approximation for $m < \sqrt{M}$ diminishes exponentially when increasing $M$, as seen by applying Chernoff bound:

$$0 \leq P(Z^{(m)} > \tau) \leq e^{-\frac{(\tau - m\mu)^2}{2m\sigma^2}} \leq e^{-\frac{(M/2 - m)^2 \mu^2}{2m\sigma^2}}$$

where $Z^{(m)} \sim \mathcal{N}(m\mu, m\sigma^2)$. Therefore, the error resulting in replacing $T_n^{(m)}$ with a Gaussian approximation, is bounded:

$$P(T_n^{(m)} < \tau) - P(Z^{(m)} < \tau) = P(Z^{(m)} > \tau) - P(T_n^{(m)} > \tau)$$

$$\Downarrow$$

$$-\frac{2m}{M} \leq -P(T_n^{(m)} > \tau) \leq P(T_n^{(m)} < \tau) - P(Z^{(m)} < \tau) \leq P(Z^{(m)} > \tau) \leq e^{-\frac{(M/2 - m)^2 \mu^2}{2m\sigma^2}}$$

$$\Downarrow$$

$$-\frac{2m}{M} + P(Z^{(m)} < \tau) \leq P(T_n^{(m)} < \tau) \leq P(Z^{(m)} < \tau) + e^{-\frac{(M/2 - m)^2 \mu^2}{2m\sigma^2}}$$

Plugging these inequalities into the left term in equation 8 gives us:

$$\sum_{m=1}^{\lfloor\sqrt{M}\rfloor} P\left(T_n^{(m)} < \tau\right) \leq \sum_{m=1}^{\lfloor\sqrt{M}\rfloor} \left(P\left(Z^{(m)} < \tau\right) + e^{-\frac{(M/2-m)^2\mu^2}{2m\sigma^2}}\right)$$

$$\tag{9}$$

$$\sum_{m=1}^{\lfloor\sqrt{M}\rfloor} P\left(T_n^{(m)} < \tau\right) \geq \sum_{m=1}^{\lfloor\sqrt{M}\rfloor} \left(P\left(Z^{(m)} < \tau\right) - \frac{2m}{M}\right) = O(1) + \sum_{m=1}^{\lfloor\sqrt{M}\rfloor} P\left(Z^{(m)} < \tau\right)$$

Combining equations 8,9 we can write the expected value as

$$\mathbb{E}[\tilde{M}(\tau)] = \sum_{m=1}^{M} P\left(T_n^{(m)} < \tau\right) = O(1) + \sum_{m=1}^{M} \Phi\left(\frac{\tau - m\cdot\mu}{\sqrt{m\cdot\sigma^2}}\right). \tag{10}$$

**Effective speedup.** As seen in section 4.4, we define the effective speedup as

$$S_{\text{eff}}(\tau) = \frac{\tilde{M}(\tau)(T + T^c)}{M \cdot (\min\{\tau, T\} + T^c)}.$$

We are interested in calculating the expected value for the effective speedup, and in order to use the formulations in equations 7,10 we first need to show that $\mathbb{E}[\tilde{M}(\tau) \cdot T] \approx \mathbb{E}[\tilde{M}(\tau)] \cdot \mathbb{E}[T]$. We examine

$$\mathbb{E}[\tilde{M}T] = \mathbb{E}[T(\mathbb{E}[\tilde{M}] + \tilde{M} - \mathbb{E}[\tilde{M}])] = \mathbb{E}[\tilde{M}]\mathbb{E}[T] + \mathbb{E}[T(\tilde{M} - \mathbb{E}[\tilde{M}])]$$

Applying Cauchy–Schwarz inequality we get

$$|\mathbb{E}[T(\tilde{M} - \mathbb{E}[\tilde{M}])]| \leq \sqrt{\mathbb{E}[T^2]\mathbb{E}[(\tilde{M} - \mathbb{E}[\tilde{M}])^2]} = \sqrt{\mathbb{E}[T^2]\frac{\sigma_{\tilde{M}_n}^2}{N}} = O(N^{-\frac{1}{2}}),$$

where $\sigma_{\tilde{M}_n}^2$ denotes the variance of $\tilde{M}_n(\tau) = \sum_{m=1}^{M} \left\{ \begin{array}{ll} 1, & \text{if } T_n^{(m)} < \tau \\ 0, & \text{otherwise} \end{array} \right\}$. Hence:

$$\mathbb{E}[\tilde{M}T] = \mathbb{E}[\tilde{M}]\mathbb{E}[T] + O(N^{-\frac{1}{2}})$$

We can now write the expected value for the effective speedup as:

$$\mathbb{E}[S_{\text{eff}}] = \frac{\sum_{m=1}^{M} \Phi\left(\frac{\tau - m \cdot \mu}{\sqrt{m\sigma^2}}\right)}{M} \cdot \frac{\mathbb{E}[T]}{\min(\tau, \mathbb{E}[T]) + T^c} + O(M^{-1} + M^{-1}N^{-\frac{1}{2}}) \tag{11}$$

$$\mathbb{E}[T] \approx \sqrt{M\sigma^2}\left((1-\gamma) \cdot \Phi^{-1}\left(1 - \frac{1}{N}\right) + \gamma \cdot \Phi^{-1}\left(1 - \frac{1}{eN}\right)\right) + M\mu + T^c \tag{12}$$

As mentioned above, when the Gaussian Assumption C.2 is inaccurate it may be useful to plug instead in the empirical value for $\mathbb{E}[T]$ in equation 11 in order to get a more accurate estimation of $\mathbb{E}[S_{\text{eff}}]$.

**Finding $\tau^*$.** The optimal threshold $\tau^*$ can be chosen as:

$$\tau^* = \text{argmax}_\tau \mathbb{E}[S_{\text{eff}}(\tau)] = \text{argmax}_\tau \left(\frac{1}{\tau + T^c} \cdot \sum_{m=1}^{M} \Phi\left(\frac{\tau - m \cdot \mu}{\sqrt{m \cdot \sigma^2}}\right)\right)$$

By using the above derivations, we can utilize $\mu$, $\sigma$, $T^c$ to understand the potential value of *Drop-Compute*. This can be done without actually training and measuring the empiric distribution of $t_n^{(m)}$ as done in appendix section C.1. We note that finding $\tau^*$ does not require any estimation of $T$ and can be done without any statistics that originate from a large scale training session.

### C.3   Additive noise analysis

As a conclusion of the previous sections, we understand that the effectiveness of *DropCompute* is mostly due to the behavior of the stochastic latency of each worker. To analyze this phenomenon we simulate a training of multiple workers using the scheme presented in section B.1 with various additive noise types. As shown in figures 13, 14, the ratio $\mathbb{E}[T]/\mathbb{E}[T_i]$ is good indicator for determining the potential of *DropCompute* on a given training setting. High ratios indicate a gap between the step time for a single worker and the step time for multiple workers, that can be compensated by using *DropCompute*.

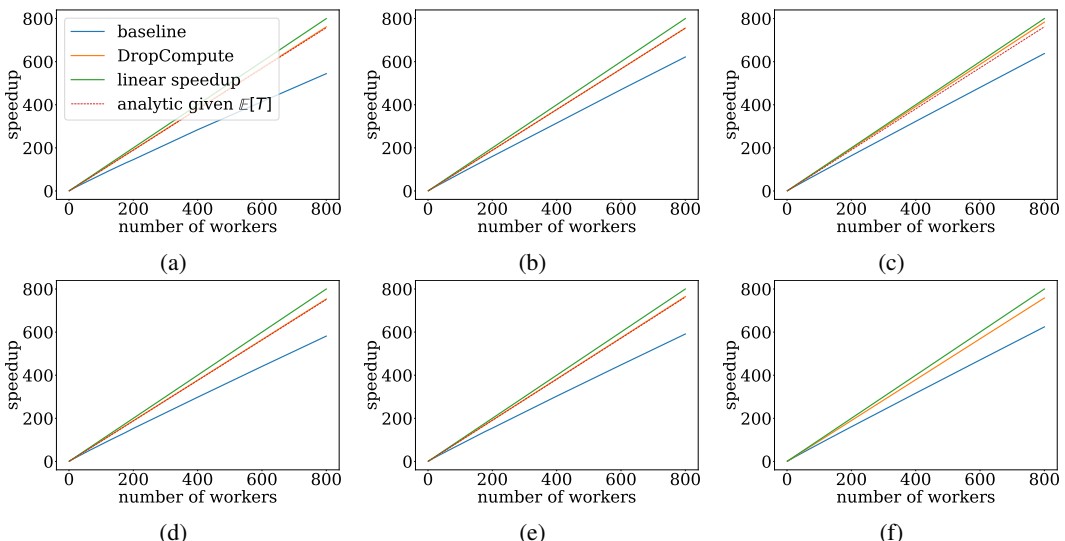

Figure 13: **The noise distribution type impacts the effectiveness of *DropCompute*.** Simulated scale graphs for a run with 12 accumulations. The duration of each accumulation is set to $0.45 + \epsilon$ seconds Different sub-graphs exhibit different distributions for $\epsilon$. The bottom-right graph was drawn using the approximation from equation 6.

| figure | Mean($\epsilon$) | Var($\epsilon$) | $\epsilon$ distribution | | $\mathbb{E}[T]/\mathbb{E}[T_i]$ |
|--------|------|------|-------------|--------------------------------|-------|
| a | 0.225 | 0.05 | lognormal | $LN(\mu = -1.84, \sigma = 0.83)$ | 1.496 |
| b | 0.225 | 0.05 | normal | $\mathcal{N}(\mu = 0.23, \sigma = 0.22)$ | 1.302 |
| c | 0.225 | 0.05 | bernoulli | $0.45Br(p = 0.5)$ | 1.283 |
| d | 0.225 | 0.05 | exponential | $exp(\lambda = 4.47)$ | 1.386 |
| e | 0.225 | 0.05 | gamma | $\gamma(\alpha = 1, \beta = 4.5)$ | 1.39 |

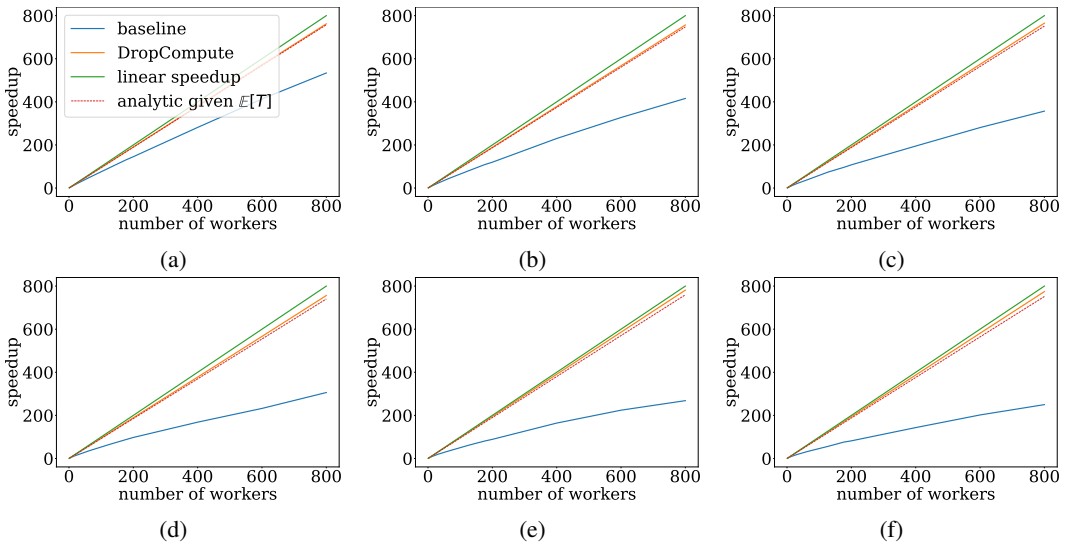

Figure 14: ***DropCompute* increases robustness to the noise variance.** Simulated scale graphs for a run with 12 accumulations. The duration of each accumulation is set to $0.45 + \epsilon$ seconds. $\epsilon$ is a stochastic lognormal random variable with a mean of $0.225$ and changing variance, according to the table below

| figure | Mean($\epsilon$) | Var($\epsilon$) | $\epsilon$ distribution | | $\mathbb{E}[T]/\mathbb{E}[T_i]$ |
|--------|------|------|-------------|--------------------------------|-------|
| a | 0.225 | 0.05 | lognormal | $LN(\mu = -1.84, \sigma = 0.83)$ | 1.496 |
| b | 0.225 | 0.1 | lognormal | $LN(\mu = -2.04, \sigma = 1.04)$ | 1.933 |
| c | 0.225 | 0.15 | lognormal | $LN(\mu = -2.18, \sigma = 1.17)$ | 2.394 |
| d | 0.225 | 0.2 | lognormal | $LN(\mu = -2.29, \sigma = 1.26)$ | 2.773 |
| e | 0.225 | 0.25 | lognormal | $LN(\mu = -2.38, \sigma = 1.33)$ | 3.043 |
| f | 0.225 | 0.3 | lognormal | $LN(\mu = -2.46, \sigma = 1.39)$ | 3.4 |

# D  Convergence with stochastic batch size

In this section, we provide proof of theorem 4.1 (D.2), as well as convergence proof for the loss itself in the convex case (D.1). We also discuss the generalization properties with a stochastic batch in section D.4.

## D.1  Proof for convex case

**Theorem D.1.** *Under assumption 4.1 and the specific case where $f$ is a convex function, for SGD with DropCompute (Algorithm 1), given $N$ workers and a local batch size $b_{local}$, we have that*

$$\mathbb{E}[\mathcal{L}(\mathcal{D}, \bar{\theta}) - \mathcal{L}(\mathcal{D}, \theta^*)] \leq \frac{8Lb_{\max}\|\theta_1 - \theta^*\|^2}{K} + \frac{6\sigma\|\theta_1 - \theta^*\|}{\sqrt{K}} \ . \tag{13}$$

*where $K$ is the total number of samples used throughout the algorithm [3], and the expectation is with respect to the randomization introduced due to sampling from $\mathcal{D}$ throughout the optimization process.*

**Proof of theorem D.1**

**Notation:** During the proof we denote by $b_i$ the total batch size (summed over all workers) that we employ at iteration $i$. $K = \sum_{i=1}^{S} b_i$ is the total number of samples along all $S$ iterations. At iteration $i$ we maintain a weight vector $\theta_i \in \mathbb{R}^d$ and query a gradient estimate $g_i$ based on a batch size of $b_i$ samples. Thus, we set

$$g_i = \frac{1}{b_i} \sum_{s=1}^{b_i} g_i^s$$

where $g_i^s = \nabla \ell(z_i^s, \theta_i)$, and $z_i^s$ is randomly sampled from $\mathcal{D}$. We also maintain importance weights, $\alpha_i = b_i$.

Note that there exists $b_{\max}$ such that $\alpha_i \leq b_{\max}$ ; $\forall i$

Now, the update rule:

$$\theta_{i+1} = \theta_i - \eta \alpha_i g_i \tag{14}$$

Eventually, we output

$$\bar{\theta} = \frac{1}{\alpha_{1:S}} \sum_{i=1}^{S} \alpha_i \theta_i$$

where $\alpha_{1:S} := \sum_{i=1}^{S} \alpha_i$.

We assume that the batch sizes $b_i$ are stopping times w.r.t. the natural filtration induced by the samples we draw during the optimization process. Informally, this means that the value of $b_i$ depends only on the history of the samples that we have seen prior to setting $b_i$.

We can therefore prove the following lemma,

**Lemma D.1.** *Upon choosing $\alpha_i = b_i$ the following holds,*

$$\mathbb{E}[\alpha_i(g_i - \nabla \mathcal{L}(\mathcal{D}, \theta_i))|\theta_i] = \mathbb{E}[\sum_{s=1}^{b_i}(g_i^s - \nabla \mathcal{L}(\mathcal{D}, \theta_i))|\theta_i] = 0 \ .$$

Now, using standard analysis for Online Gradient Descent (Hazan et al., 2016) with the update rule of (14) gives,

$$\sum_{i=1}^{S} \alpha_i g_i \cdot (\theta_i - \theta^*) \leq \frac{\|\theta_1 - \theta^*\|^2}{\eta} + \eta \sum_{i=1}^{S} \alpha_i^2 \|g_i\|^2$$

Taking expectation and using Lemma D.1 gives,

$$\mathbb{E} \sum_{i=1}^{S} \alpha_i \nabla \mathcal{L}(\mathcal{D}, \theta_i) \cdot (\theta_i - \theta^*) \leq \frac{\|\theta_1 - \theta^*\|^2}{\eta} + \eta \mathbb{E} \sum_{i=1}^{S} \alpha_i^2 \|g_i\|^2$$

---

[3]We assume that the batch sizes may be stochastic but $K$ is predefined and deterministic

From convexity we know that $0 \leq \mathcal{L}(\mathcal{D}, \theta_i) - \mathcal{L}(\mathcal{D}, \theta^*) \leq \nabla \mathcal{L}(\mathcal{D}, \theta_i) \cdot (\theta_i - \theta^*)$, therefore the above implies,

$$\mathbb{E} \sum_{i=1}^{S} \alpha_i (\mathcal{L}(\mathcal{D}, \theta_i) - \mathcal{L}(\mathcal{D}, \theta^*)) \leq \frac{\|\theta_1 - \theta^*\|^2}{\eta} + \eta \mathbb{E} \sum_{i=1}^{S} \alpha_i^2 \|g_i\|^2 \qquad (15)$$

Now, we write $g_i = \nabla \mathcal{L}(\mathcal{D}, \theta_i) + \xi_i$ where $\xi_i = g_i - \nabla \mathcal{L}(\mathcal{D}, \theta_i)$ and note that

$$\alpha_i \xi_i = \sum_{s=1}^{b_i} (g_i^s - \nabla \mathcal{L}(\mathcal{D}, \theta_i)) = \sum_{s=1}^{b_i} \xi_i^i$$

where we denote $\xi_i^s := g_i^s - \nabla \mathcal{L}(\mathcal{D}, \theta_i)$. Next, we shall use the following lemma,

**Lemma D.2.** *The following holds,*

$$\mathbb{E} \alpha_i^2 \|g_i\|^2 \leq 2 b_{\max} \mathbb{E} \alpha_i \|\nabla \mathcal{L}(\mathcal{D}, \theta_i)\|^2 + 2\sigma^2 \mathbb{E} b_i . \qquad (16)$$

*Moreover, due to the $L$-smoothness of $f(\cdot)$, and global optimality of $\theta^*$, the following holds,*

$$\mathbb{E} \alpha_i^2 \|g_i\|^2 \leq 4 b_{\max} L \mathbb{E} \alpha_i (\mathcal{L}(\mathcal{D}, \theta_i) - \mathcal{L}(\mathcal{D}, \theta^*)) + 2\sigma^2 \mathbb{E} b_i . \qquad (17)$$

**Final Bound:** Plugging the above lemma back into Eq. (15) gives,

$$\mathbb{E} \sum_{i=1}^{S} \alpha_i (\mathcal{L}(\mathcal{D}, \theta_i) - \mathcal{L}(\mathcal{D}, \theta^*)) \leq \frac{\|\theta_1 - \theta^*\|^2}{\eta} + 4\eta b_{\max} \mathbb{E} \sum_{i=1}^{S} \alpha_i L(\mathcal{L}(\mathcal{D}, \theta_i) - \mathcal{L}(\mathcal{D}, \theta^*)) + 8\eta\sigma^2 \mathbb{E} \sum_{i=1}^{S} b_i$$

$$\leq \frac{\|\theta_1 - \theta^*\|^2}{\eta} + 4\eta b_{\max} L \mathbb{E} \sum_{i=1}^{S} \alpha_i (\mathcal{L}(\mathcal{D}, \theta_i) - \mathcal{L}(\mathcal{D}, \theta^*)) + 8\eta\sigma^2 K ,$$

$$(18)$$

where we used $K = \sum_{i=1}^{S} b_i$.

Now if we pick $\eta$ such that $4\eta b_{\max} L \leq 1/2$ then we can move the second term in the RHS to the LHS and obtain,

$$\frac{1}{2} \mathbb{E} \sum_{i=1}^{S} \alpha_i (\mathcal{L}(\mathcal{D}, \theta_i) - \mathcal{L}(\mathcal{D}, \theta^*)) \leq \frac{\|\theta_1 - \theta^*\|^2}{\eta} + 8\eta\sigma^2 K , \qquad (19)$$

Thus, choosing $\eta = \min \left\{ \frac{\|\theta_1 - \theta^*\|}{\sigma \sqrt{8K}}, \frac{1}{8Lb_{\max}} \right\}$ gives the following bound,

$$\mathbb{E} \sum_{i=1}^{S} \alpha_i (\mathcal{L}(\mathcal{D}, \theta_i) - \mathcal{L}(\mathcal{D}, \theta^*)) \leq 8Lb_{\max} \|\theta_1 - \theta^*\|^2 + 6\sigma \|\theta_1 - \theta^*\| \sqrt{K} \qquad (20)$$

Now, recalling that $K := \sum_{i=1}^{S} b_i = \sum_{i=1}^{S} \alpha_i$ and using Jensen's inequality together with $\bar{\theta} = \frac{1}{\alpha_{1:S}} \sum_{i=1}^{S} \alpha_i \theta_i$ yields,

$$\mathbb{E}[\mathcal{L}(\mathcal{D}, \bar{\theta}) - \mathcal{L}(\mathcal{D}, \theta^*)] \leq \mathbb{E} \sum_{i=1}^{S} \frac{\alpha_i}{\alpha_{1:S}} (\mathcal{L}(\mathcal{D}, \theta_i) - \mathcal{L}(\mathcal{D}, \theta^*)) \leq \frac{8Lb_{\max} \|\theta_1 - \theta^*\|^2 + 6\sigma \|\theta_1 - \theta^*\| \sqrt{K}}{\alpha_{1:S}}$$

$$\leq \frac{8Lb_{\max} \|\theta_1 - \theta^*\|^2}{K} + \frac{6\sigma \|\theta_1 - \theta^*\|}{\sqrt{K}} \qquad (21)$$

where we used $\alpha_{1:S} = K$. $\quad \square$

### D.2 Proof for non-convex case

**Proof of theorem 4.1**

We use the same notation for $b_i$ and $g_i$ as before. And again used weights,

$$\alpha_i = b_i$$

We also assume that $b_i \leq b_{\max}$ , $\forall i$.

The update rule is the following,

$$\theta_{i+1} = \theta_i - \eta\alpha_i g_i$$

And the output is $\bar{\theta}$, where we define,

$$\bar{\theta} = \theta_i \; ; \quad \text{w.p.} \;\; \frac{\alpha_i}{\alpha_{1:S}}$$

Thus,

$$\mathbb{E}\|\nabla\mathcal{L}(\mathcal{D},\bar{\theta})\|^2 = \frac{1}{\alpha_{1:S}}\sum_{i=1}^{S}\alpha_i\mathbb{E}\|\nabla\mathcal{L}(\mathcal{D},\theta_i)\|^2$$

Using smoothness,

$$\mathcal{L}(\mathcal{D},\theta_{i+1}) \leq \mathcal{L}(\mathcal{D},\theta_i) - \nabla\mathcal{L}(\mathcal{D},\theta_i) \cdot (\theta_{i+1} - \theta_i) + \frac{L}{2}\|\theta_{i+1} - \theta_i\|^2$$

$$\leq \mathcal{L}(\mathcal{D},\theta_i) - \eta\alpha_i\nabla\mathcal{L}(\mathcal{D},\theta_i) \cdot g_i + \frac{L\eta^2}{2}\|\alpha_i g_i\|^2$$

$$\leq \mathcal{L}(\mathcal{D},\theta_i) - \eta\alpha_i\|\nabla\mathcal{L}(\mathcal{D},\theta_i)\|^2 - \eta\alpha_i\nabla\mathcal{L}(\mathcal{D},\theta_i) \cdot (g_i - \nabla\mathcal{L}(\mathcal{D},\theta_i)) + \frac{L\eta^2}{2}\|\alpha_i g_i\|^2$$

Re-arranging the above yields,

$$\eta\alpha_i\|\nabla\mathcal{L}(\mathcal{D},\theta_i)\|^2 \leq \mathcal{L}(\mathcal{D},\theta_i) - \mathcal{L}(\mathcal{D},\theta_{i+1}) - \eta\alpha_i\nabla\mathcal{L}(\mathcal{D},\theta_i) \cdot (g_i - \nabla\mathcal{L}(\mathcal{D},\theta_i)) + \frac{L\eta^2}{2}\|\alpha_i g_i\|^2$$

Summing the above, and dividing by $\eta$, we obtain,

$$\sum_{i=1}^{S}\alpha_i\|\nabla\mathcal{L}(\mathcal{D},\theta_i)\|^2 \leq \frac{1}{\eta}(\mathcal{L}(\mathcal{D},\theta_1) - \mathcal{L}(\mathcal{D},\theta_{S+1})) - \sum_{i=1}^{S}\alpha_i\nabla\mathcal{L}(\mathcal{D},\theta_i) \cdot (g_i - \nabla\mathcal{L}(\mathcal{D},\theta_i)) + \frac{L\eta}{2}\sum_{i=1}^{S}\|\alpha_i g_i\|^2$$

$$\leq \frac{1}{\eta}(\mathcal{L}(\mathcal{D},\theta_1) - \mathcal{L}(\mathcal{D},\theta^*)) - \sum_{i=1}^{S}\alpha_i\nabla\mathcal{L}(\mathcal{D},\theta_i) \cdot (g_i - \nabla\mathcal{L}(\mathcal{D},\theta_i)) + \frac{L\eta}{2}\sum_{i=1}^{S}\|\alpha_i g_i\|^2 \tag{22}$$

where we uses $\mathcal{L}(\mathcal{D},\theta^*) \leq \mathcal{L}(\mathcal{D},\theta_{S+1})$ since $\theta^*$ is the global minimum of $\mathcal{L}(\mathcal{D},\cdot)$.

Now recall that from Lemma D.1 we have,

$$\mathbb{E}[\alpha_i(g_i - \nabla\mathcal{L}(\mathcal{D},\theta_i))|\theta_i] = 0 . \tag{23}$$

And from Lemma D.2 we have,

$$\mathbb{E}\alpha_i^2\|g_i\|^2 \leq 2b_{\max}\mathbb{E}\alpha_i\|\nabla\mathcal{L}(\mathcal{D},\theta_i)\|^2 + 2\sigma^2\mathbb{E}b_i . \tag{24}$$

Thus, taking expectation in Eq. (22), and plugging Eq. (23) and (24), yields,

$$\mathbb{E}\sum_{i=1}^{S}\alpha_i\|\nabla\mathcal{L}(\mathcal{D},\theta_i)\|^2 \leq \frac{1}{\eta}(\mathcal{L}(\mathcal{D},\theta_1) - \mathcal{L}(\mathcal{D},\theta^*)) - \sum_{i=1}^{S}\mathbb{E}\alpha_i\nabla\mathcal{L}(\mathcal{D},\theta_i) \cdot (g_i - \nabla\mathcal{L}(\mathcal{D},\theta_i)) + \frac{L\eta}{2}\sum_{i=1}^{S}\mathbb{E}\|\alpha_i g_i\|^2$$

$$\leq \frac{1}{\eta}(\mathcal{L}(\mathcal{D},\theta_1) - \mathcal{L}(\mathcal{D},\theta^*)) + L\eta b_{\max}\sum_{i=1}^{S}\mathbb{E}\alpha_i\|\nabla\mathcal{L}(\mathcal{D},\theta_i)\|^2 + L\eta\sigma^2\mathbb{E}\sum_{i=1}^{S}b_i$$

$$\leq \frac{1}{\eta}(\mathcal{L}(\mathcal{D},\theta_1) - \mathcal{L}(\mathcal{D},\theta^*)) + L\eta b_{\max}\sum_{i=1}^{S}\mathbb{E}\alpha_i\|\nabla\mathcal{L}(\mathcal{D},\theta_i)\|^2 + L\eta\sigma^2 \cdot K \tag{25}$$

where the last line uses $K := \sum_{i=1}^{S}b_i$.

Now if we pick $\eta$ such that $\eta b_{\max} L \leq 1/2$ then we can move the second term in the RHS to the LHS and obtain,

$$\frac{1}{2}\mathbb{E}\sum_{i=1}^{S}\alpha_i\|\nabla\mathcal{L}(\mathcal{D},\theta_i)\|^2 \leq \frac{1}{\eta}(\mathcal{L}(\mathcal{D},\theta_1) - \mathcal{L}(\mathcal{D},\theta^*)) + L\eta\sigma^2 \cdot K \tag{26}$$

Thus, choosing $\eta = \min\left\{\frac{\sqrt{\mathcal{L}(\mathcal{D},\theta_1)-\mathcal{L}(\mathcal{D},\theta^*)}}{\sigma\sqrt{LK}}, \frac{1}{2Lb_{\max}}\right\}$ gives the following bound,

$$\mathbb{E}\sum_{i=1}^{S}\alpha_i\|\nabla\mathcal{L}(\mathcal{D},\theta_i)\|^2 \leq 2Lb_{\max}(\mathcal{L}(\mathcal{D},\theta_1) - \mathcal{L}(\mathcal{D},\theta^*)) + 2\sigma\sqrt{L(\mathcal{L}(\mathcal{D},\theta_1) - \mathcal{L}(\mathcal{D},\theta^*))} \cdot \sqrt{K}$$

$$\tag{27}$$

Dividing by $K := \alpha_{1:S}$ and using the definition of $\bar{\theta}$ yields,

$$\mathbb{E}\|\nabla\mathcal{L}(\mathcal{D},\bar{\theta})\|^2 = \mathbb{E}\frac{1}{\alpha_{1:S}}\sum_{i=1}^{S}\alpha_i\|\nabla\mathcal{L}(\mathcal{D},\theta_i)\|^2 \leq \frac{2Lb_{\max}(\mathcal{L}(\mathcal{D},\theta_1) - \mathcal{L}(\mathcal{D},\theta^*))}{K} + \frac{2\sigma\sqrt{L(\mathcal{L}(\mathcal{D},\theta_1) - \mathcal{L}(\mathcal{D},\theta^*))}}{\sqrt{K}}$$

$$\tag{28}$$

### D.3   Remaining Proofs

#### D.3.1   Proof of Lemma D.2

*Proof of Lemma D.2.* We can write,

$$\alpha_i^2\|g_i\|^2 = \|b_i\nabla\mathcal{L}(\mathcal{D},\theta_i) + \sum_{s=1}^{b_i}\xi_i^s\|^2$$

$$\leq 2\|b_i\nabla\mathcal{L}(\mathcal{D},\theta_i)\|^2 + 2\|\sum_{s=1}^{b_i}\xi_i^s\|^2$$

$$= 2b_i^2\|\nabla\mathcal{L}(\mathcal{D},\theta_i)\|^2 + 2\|\sum_{s=1}^{b_i}\xi_i^s\|^2$$

$$\leq 2b_{\max}\alpha_i\|\nabla\mathcal{L}(\mathcal{D},\theta_i)\|^2 + 2\|\sum_{s=1}^{b_i}\xi_i^s\|^2$$

$$\tag{29}$$

where the second line uses $\|a + b\|^2 \leq 2\|a\|^2 + 2\|b\|^2$; the fourth line uses $b_i \leq b_{\max}$ as well as $\alpha_i = b_i$ implying that $b_i^2 \leq b_{\max}\alpha_i$.

**Bounding $E\|\sum_{s=1}^{b_i}\xi_i^s\|^2$:**   Given $i$ and $\theta_i$, Let us define the following sequence, $Q_0 = 0$, $Q_1 = \|\xi_i^1\|^2 - \sigma$, and for any $k > 1$

$$Q_k = \sum_{s=1}^{k}\|\xi_i^k\|^2 - \sigma^2 \cdot k + 2\sum_{s=1}^{j}\sum_{n=s+1}^{j}\xi_i^s \cdot \xi_i^n$$

It can be directly shown that $\{Q_k\}_k$ is a Supermartingale sequence, and that

$$\|\sum_{s=1}^{b_i}\xi_i^s\|^2 = Q_{b_i} + \sigma^2 \cdot b_i$$

Thus, since $b_i$ is a bounded stopping time, we can use Doob's optional stopping theorem which implies that,

$$\mathbb{E}\|\sum_{s=1}^{b_i}\xi_i^s\|^2 = \mathbb{E}Q_{b_i} + \sigma^2\mathbb{E}\cdot b_i \leq EQ_0 + \sigma^2\mathbb{E}\cdot b_i = 0 + \sigma^2\mathbb{E}b_i$$

Plugging the above back into Eq. (29) yields,

$$\mathbb{E}\alpha_i^2\|g_i\|^2 \leq 2b_{\max}\mathbb{E}\alpha_i\|\nabla\mathcal{L}(\mathcal{D},\theta_i)\|^2 + 2\sigma^2\mathbb{E}b_i \ . \tag{30}$$

Now, since $\mathcal{L}(\mathcal{D},\cdot)$ is $L$-smooth and $\theta^*$ is its global minima, then the following holds: $\|\nabla\mathcal{L}(\mathcal{D},\theta_i)\|^2 \leq 2L(\mathcal{L}(\mathcal{D},\theta_i) - \mathcal{L}(\mathcal{D},\theta^*))$; See e.g. Levy (2017) for the proof. Plugging this into the above equation we obtain,

$$\mathbb{E}\alpha_i^2\|g_i\|^2 \leq 4b_{\max}\mathbb{E}\alpha_i L(\mathcal{L}(\mathcal{D},\theta_i) - \mathcal{L}(\mathcal{D},\theta^*)) + 2\sigma^2\mathbb{E}b_i \ . \tag{31}$$

$\square$

### D.3.2 Proof of Lemma D.1

*Proof of Lemma D.1.* We can define the following Martingale sequence for each step $s$: $M_0 = 0$, and $M_j = \sum_{s=1}^{j}(g_i^s - \nabla\mathcal{L}(\mathcal{D},\theta_i))$ for any $j = 1, 2, \ldots$.

Thus, since the mixing time $b_i$ is bounded by $b_{\max}$, then according to Doob's optional stopping theorem (Levin & Peres, 2017) we have that,

$$\mathbb{E}[M_{b_i}|\theta_i] = \mathbb{E}[\sum_{s=1}^{b_i}(g_i^s - \nabla\mathcal{L}(\mathcal{D},\theta_i))|\theta_i] = \mathbb{E}[M_0|\theta_i] = 0 \ . \tag{32}$$

Now, notice that for any $i$, we have $\alpha_i(g_i - \nabla\mathcal{L}(\mathcal{D},\theta_i)) = M_{b_i}$, and therefore,

$$\mathbb{E}[\alpha_i(g_i - \nabla\mathcal{L}(\mathcal{D},\theta_i))|\theta_i] = \mathbb{E}[M_{b_i}|\theta_i] = 0 \ . \tag{33}$$

$\square$

### D.4 Generalization discussion

An interesting observation arising from our results is the small impact of gradient dropping as measured in final test accuracy. One explanation for this can be based on viewing *DropCompute* as noise induced over gradients. Optimization using variants of SGD is inherently noisy due to the use of data samples used to evaluate the intermediate error. The stochastic nature of computed weight gradients was previously found to provide generalization benefits for the final trained model, although this is still part of ongoing debate (Geiping et al., 2022). Nevertheless, several works found generalization benefits with the *injection* of noise into the weights' gradients (Neelakantan et al., 2015) or their use in computed update rule (Lin et al., 2022).