# OpenReview forum: "DropCompute: simple and more robust  distributed synchronous training via compute variance reduction"
_NeurIPS.cc/2023/Conference — NeurIPS 2023 poster_

### Official Review · Reviewer_PA5g · 2023-06-23

**Soundness:** 2 fair
**Presentation:** 3 good
**Contribution:** 3 good
**Rating:** 6
**Confidence:** 2

**Summary:**

This paper considers a typical scenario in which workers are straggling due to variability in compute time, in which the authors find an analytical relation between compute time properties and scalability limitations. They propose a decentralized method to reduce the variation among workers and   improve the robustness of synchronous training.

**Strengths:**

1. This paper studies the distribution of iteration time and presents some insights into understanding distributed optimization.

2. Based on the analysis of the iteration time for the worker, the authors propose an algorithm that drops some data in computing local gradients to avoid waiting.

3. They present the analysis of distributed SGD with stochastic batch size, which is novel and interesting.

4. Numerical results are presented to verify the assumptions and demonstrate that the robustness of the convergence.

**Weaknesses:**

1. The convergence is proved under many extra assumptions than that of distributed SGD and cannot explain why the new algorithm can be better.

2. According to Figure 5, the proposed algorithm seems only a little better than the baseline.

**Questions:**

What is the computation cost of Algorithm 2? Does it consume much time, and is the consuming time recorded in  Figure 5?

**Limitations:**

Yes

---

> ### Author Rebuttal · Authors · 2023-08-09
>
> ### Weakness 1 response
> We prove convergence under the standard assumptions that are used in the analysis of distributed SGD. To be concrete, we make 2 assumptions: Smoothness, and bounded variance, both of them are standard in the analysis of distributed/parallel SGD, see e.g. the classical paper of [*Dekel et al, 201*2]
>
> We prove results for both the non-convex case (in Theorem 4.1) and Convex case (in Theorem C.1)
>
> Lastly, to remove any doubt, the unbiasedness assumption that we mention stems directly from the property of stochastic gradients.
>
>
> ### Weakness 2 response
> We respectfully disagree, see (**GQ2: How effective is DropCompute in practice?**).
>
> ### Question 1 response
> The computation cost of algorithm 2 is $\\theta(N\\cdot I\\cdot M)$ where $N$ is the number of workers, $I$ is the number of recorded iterations and $M$ is the number of accumulations per iteration. Overall, this cost is negligible, compared to a full training session, because it happens only once in a training session.

---

### Official Review · Reviewer_btsj · 2023-06-26

**Soundness:** 3 good
**Presentation:** 3 good
**Contribution:** 2 fair
**Rating:** 6
**Confidence:** 4

**Summary:**

The authors proposed a method to drop computation (DropCompute) of **microbatches** of a **minibatch** on decentralized workers if their execution time exceeds a certain threshhold. The author provided theoretical convergence proof of this method as well as theoretical/experimental study of the performance gain or speed up of this method.

**Strengths:**

* The proposed method is quite simple and straightforward in implementation, especially for decentralized SPMD distributed system.

* The theoretical study of the convergence and speed up of this method is thorough and clear

* The ablation study of the experimental section is clear

**Weaknesses:**

* If my understand is correct, the theoretical analysis (line 186) and efficacy (Figure 4 right) of the method relies on large enough minibatch size per replica, so that number of microbatches M >> 1. This probably worked fine for BERT training (line 241), however for modern LLM pretrainings, e.g., GPT3 XL (similar size as the model studied in this work), the global batchsize is 1M tokens/(2K tokens/sample) = 500 samples. Distributed on 200 replicas, minibatch / replica is only 2-3 samples, which fundamentally breaks the assumption. The global batchsize can be even smaller for fine-tuning. Addtionally, the evaluation on Image task (line 229) basically has only 1 microbath per minibatch, and does not serve a good experimental study for DropCompute (which is more like Federated Learning).

* The proposed method drops data/samples per iteration. This might be fine if the data is trained multiple epochs like BERT scale, however for modern LLM, especially pretraining, the data is usually trained one epoch so if one micrabatch is dropped, it might be dropped from training forever.

* The performance gain is marginal, e.g, theoretically at most 20% (Figure 3) by choosing the best threshold $\tau$ or droprate, and in practice 5% (Figure 4) if maintaining final model quality. This is likely due to the compute variance of replicas follow a normal distribution (with small std) and the system w/o DropCompute is not significantly slowed down in this case. Things can be more interesting if the variance follows an extreme distribution, e.g., normal + delta function, but that's not likely true for datacenter clusters.

* The author said no additional hyper-parameters are introduced (line 55), however proper droprate or threshold time $\tau$ is still a new hyperparameter

**Questions:**

* in step 6 Algorithm 1 the authors mentioned step 7 and step 10 are in parallel, does it mean step 9 could stop in the middle if time out in step 10 or step 8-9 are still atomic? The description seems to suggest it is the former one (otherwise step 10 can be just a break condition in the forloop), but this results in gradients with i or i - 1 steps accumulation and breaks the theoretical analysis if my understand is correct.

* is it possible to reuse dropped data/samples, e.g, there could be a synchronization of dropped sample indices of the dataloader of each replica so that next batch can be sampled with previously dropped data? Alternatively, can you cache the dropped data/samples locally on each replica, and use them in next batch. Both seem to be simple change to dataloaders.

**Limitations:**

The authors didn't talk about their limitations. But from my understand, the main limitations include the first 3 points mentioned in Weaknesses:
* large number of microbatches assumption
* training data dropping
* the problem solved is not significiant for LLM trainings which mostly happen in datacenter (compared to Federated Learning)

Additionally, the method is only applicable to traditional data parallel training, for modern distributed system that adopts FSDP/DeepSpeed-Zero and Tensor/Pipeline Parallellsim, the proposed method can't help with their parallel compute variance (because it depends on gradient accumulation, again, multiple microbatches in a minibatch).

---

> ### Author Rebuttal · Authors · 2023-08-09
>
> ### Weakness 1 response
> It is true that some learning tasks do not use many microbatches, however, we argue that for LLMs the number of microbatches can be significantly larger. For example, the recent leading performance benchmark for LLMs, GPT-3 175B training with 3584 workers (including pipeline, tensor and data parallelism), utilizes 24 micro-batches [*MLCommons MLPerf v3.0 training benchmark*]. For reference, as can be seen in our experiments (section 5.2, figure 4a), it is sufficient to use 16 microbatches.
>
> For ResNet50 experiments, the evaluation indeed uses a single microbatch, since we followed a realistic training regime. The purpose of this experiment is to diversify the learning domains our evaluations are based on, at the expense of simulation of the drops. It is possible to implement DropCompute in layers granularity (with backward hooks for example) such that workers can stop during backward pass - when reaching a timeout, stop the backprop and allreduce the gradients accumulated so far. However, this probably requires a lower level implementation to provide real value in terms of runtime performance.
>
> Nonetheless, we thank the reviewer for raising this point, and we will add it to the limitations section.
>
>
> ### Weakness 2 response
> Our evaluations consider the need to compensate for the dropped data samples with more steps. This is integrated in the definition of the "effective speedup" metric. Regarding missing samples (in case of training a single epoch), it is possible to reuse them, similar to the suggestion in your second question - saving the indices of the dropped samples and adding them back to the data loader.
>
> ### Weakness 3 response
> As written in (**GQ2: How effective is DropCompute in practice?**), the theoretical performance gain can be greater than 20% since it depends on the noise statistics (see appendix section B.3), scale, and the level of system optimizations. Moreover, even a 5-20% improvement can be significant in large-scale implementations.
>
> ### Weakness 4 response
> We note that the drop rate $\tau$ is automatically determined by the method itself for maximum effective speedup, as detailed in Section 4.4, "Choosing the Threshold". By stating that “we do not introduce additional hyper-parameters”, we mean that no additional hyper parameter tuning is required which is the main drawback of using many hyper-parameters.
>
> ### Question 1 response
> In our implementation, step 8 can be interrupted (either before forward pass or before backward pass) while step 9 is atomic - i.e., either we accumulate the gradients from the micro-batch ‘m’, or drop it entirely. This is inline with our definition of $\\tilde{M}(\\tau)$ where a sample is counted only if $T_n^{(m)} < \\tau$ and therefore, our theoretical analysis remains valid.  We thank the reviewer for raising this question and will clarify it in the paper.
> One might propose combining steps 8+9 and having the option of breaking computation during the backward pass, after calculating a portion of the micro-batch gradients. This approach is possible, and we tried it during the early stages of our experiments. According to our empirical experiments this does not harm convergence. However, for ease of use and analysis of our method, we decided to drop this idea. We thank the reviewer in seeing value in such optimizations and will include this approach in our “future directions” section.
>
>
> ### Question 2 response
> In workloads where a single epoch is trained this idea could be valuable. We thank the reviewer for this suggestion, and we will integrate it into the paper.
>
> ### Limitations response
> - training data dropping - can be solved easily using your suggestion, of saving the indices of dropped examples
>
> - problem not significant in datacenter - we respectfully disagree - (see **GQ1: Is compute variance really an issue?**).
>
> - Tensor/Pipeline Parallelism - simple DropCompute can always be used at the data-parallel level (dropping micro-batches of an entire pipeline group). For example, training GPT3 at large scale (as done by [*NVidia in MLCommons MLPerf v3.0 training benchmark*]) utilizes a model-parallel level of 32 workers, and 112 worker data-parallel (overall 3584 workers). DropCompute could be used here by dropping a straggling 32-worker model-parallel unit. Also, it is plausible to assume that the communication required in the model-parallel unit would increase the compute variance and make DropCompute even more effective. In our terms, this case has N=112 droppable workers.
> We thank the reviewer for raising this point and we will address this as part of our “limitations” and “future work” section.

---

> > ### Comment · Reviewer_btsj · 2023-08-14
> >
> > Thanks for the detailed explanations and the addtitional data on compute variance. I totally agree that a (**free and robust**) 5% speedup is siginificant in LLM trainings, however my concern is DropCompute is less applicable to LLM trainings due to:
> >
> > 1. The batch-size can indeed be larger, at the cost of developing new ML algorithms (like Optimizers) and collecting better datasets, and can hardly be applicable to fine-tuning. The 5-20% speedup is not **robust** to one of the most immportant hyper-parameters: batch-size, thus comes with a system/acc tradeoff. Additionally, LLMs are using much longer sentence size in pretraining (from 2K tokens to 4K/8K), which also limits mini batch size per replica.
> >
> > 2. Larger drop rate (>10%) comes at the loss of model accuracy (Table 1), which means it is not a **free**. ML researchers have to pick a drop rate carefully so that it improves throughputs while maintaining final model acc. That's why I brought up that this drop rate is an additional hyperarameter (as dicussed in Weakness 4 response, it can be auto inferred to maximize speed up, but maximized speed up does not guarantee final model acc). Since LLM has tons of downstream tasks depend on it, 5-10% speed up is not worth any loss in acc.
> >
> > Nonetheless, I do believe DropCompute shows clear values in training workloads with 1. large compute variance, 2. enough batch-size and 3. less downstream task dependencies due to its simplicity, just that its value in LLM training is not convincing enough to me. I'm not sure if the authors have time to experiment Question 2 response and see if it can recovers some acc loss at higher drop-rate, if so I am open to a borderline accept (5).

---

> > > ### Author Response · Authors · 2023-08-16
> > >
> > > We thank the reviewer for the positive comments on the value of DropCompute, and for the additional feedback. Regarding the remaining concerns:
> > >
> > > **1a. Is it a limitation that DropCompute requires large batch sizes?**
> > >
> > > Large batch size is not the main constraint, but the number of micro-batches used (as the reviewer pointed out in the previous comment). We focused on applying DropCompute on workloads, such as LLMs pretraining, that already require large batch sizes, and enough micro-batches (>16, as we explained before). In addition, we believe that the trend in training larger and larger LLMs is leading towards an increased number of micro batches ― a trend that can only further help DropCompute. For example, we mentioned GPT3 which had 24 microbatches (in our rebuttal), and we think this number typically ‌increases in even larger models, such as the 160 microbatches in  *[Smith et al. “Using DeepSpeed and Megatron to Train Megatron-Turing NLG 530B, A Large-Scale Generative Language Model”]*; note the authors there also explicitly mention gradient accumulation as a desired method in LLM training. Therefore, we do not believe this is a serious limitation.
> > >
> > > **1b. Are longer sentence sizes an issue for DropCompute?**
> > >
> > > No. Like the reviewer mentioned, LLMs are using longer sentence sizes in pretraining. However, we are not sure why this is a problem. In fact, this seems to benefit DropCopmute: longer sentence sizes lead to smaller micro-batches (because of memory constraints) which lead to more micro-batches per worker.
> > >
> > > **2. Does the DropCompute speedup we showed come for 'free’ (i.e. without accuracy degradation)?**
> > >
> > > Yes. There might be some misunderstanding regarding how we define the speedup versus the results in Table 1, so we would like to clarify it.
> > > Effective speedup (as defined in equation (6)) already considers the dropped samples. For example, if the drop rate is 10% and the effective speedup is 10%, it means that we have gone through ~11% more samples (while dropping 10% of all samples), in 10% less time. To compensate for the dropped samples and competitive runtime performance, we suggested doing more training steps (e.g., lines 275-277: “...achieving the same training loss as the baseline might requires additional training steps, however, it leads to a notable reduction in overall training time.”). We found this to be a hassle-free method in our experiments, as we illustrated in Figure 5. There, we trained with DropCompute (with 6.7% drop rate) until reaching the baseline training loss (instead of training a fixed budget of steps). This required 3% more steps, but took 13% less time.
> > > In Table 1, however, we report the model accuracy without adding more training steps (which leads to a further increased speedup, higher than the “effective speedup”). Therefore, we expected Table 1 to have accuracy losses. The point of Table 1 (and Figure 9 in the appendix) was to show that the actual speedup can be higher than the effective speedup, since we do not even have to add more steps, unless drop rate > 10% (we note we did not observe experiments with drop rate > 7%, even in our most extreme cases, presented in the PDF attached to the global response part of the rebuttal). If we were to add more iterations to the training sessions used to produce in Table 1 (to compensate for the drop rate), we would have reached a similar accuracy as the baseline (as we meant to say on lines 251-252, we apologize if this was not very clear).
> > >
> > > We hope that this answers any remaining concerns. Please let us know if there are any other concerns.

---

> > > > ### Comment · Reviewer_btsj · 2023-08-16
> > > >
> > > > Thank you again for your response.
> > > >
> > > > > However, we are not sure why this is a problem.
> > > >
> > > > In LLM training, the batch-size is actually num-tokens rather than num-sentences, when num-tokens is fixed, the longer the sentence, the smaller the num of sentences per replica.
> > > >
> > > > > Effective speedup ... in 10% less time.
> > > >
> > > > Thanks for clarifying, I might miss this detail in the context. Given that I am willing to increase the score to borderline 4. If you could conduct experiments with a compensating dataloader instead of dropping samples + more training steps, as I mentioned earlier I am open to a weak accept 6.

---

> > > > > ### Author Response · Authors · 2023-08-21
> > > > >
> > > > > We thank the reviewer for the additional feedback.
> > > > >
> > > > > As requested, we conducted additional experiments with the settings described in Table 1, for BERT-Large pretraining. We focused on the most challenging case in Table 1, of 10% droprate. There, we had some degradation in accuracy (F1 score): from 91.32 +- 0.15 in the baseline without DropCompute, to 91.13 +- 0.02 with 10% droprate and without compensating at all for the dropped samples.
> > > > >
> > > > > We have tested 3 methods of compensating for the dropped samples:
> > > > > 1. Additional steps: training for extra ~11% steps (to maintain, on average, the total number of samples seen during training without DropCompute). This resulted in the baseline accuracy (91.40 +- 0.08). This validates our suggested approach of taking extra steps into account.
> > > > > 2. Increased batch size: training with an increased global batch size of ~11% (before dropping), to maintain, on average, the same batch size as the baseline (after dropping). This approach also resulted in the baseline performance (91.38 +- 0.08).
> > > > > 3. Saving the dropped samples indices, and reusing them before starting a new epoch. As suggested by the reviewer, this method uses the dropped samples explicitly. We have made the necessary code alterations to the data loader, but haven't finished yet to validate and complete the pretraining experiment. This experiment will not be completed by the end of the discussion period, but will be included in the final submission along with the first two results.
> > > > >
> > > > > We note that the first two methods do not reuse the dropped samples explicitly, yet the accuracy does not degrade. Therefore, there is no remaining gap to close by using the third approach (reusing he dropped samples), but it may accelerate training further (perhaps by combining with the other approaches). It also may be beneficial to accuracy in even more challenging cases. We thank the reviewer (and also reviewer 5jtg) for suggesting this valuable improvement of our method. We will definitely include it in the paper.

---

> > > > > > ### Comment · Reviewer_btsj · 2023-08-21
> > > > > >
> > > > > > Thanks for the quick updates. I have raised my score to 6 and looking forward to your final results.

---

### Official Review · Reviewer_5jtg · 2023-07-07

**Soundness:** 3 good
**Presentation:** 3 good
**Contribution:** 2 fair
**Rating:** 6
**Confidence:** 4

**Summary:**

This paper introduces DropCompute, a method to mitigate load imbalance across workers when doing distributed training. It splits mini-batches into micro-batches, and uses a dynamically-calculated time budget, which, when exceeded, results in the remaining micro-batches being dropped and communication to aggregate gradients beginning. A convergence analysis is provided, and experiments are conducted on ResNet-50 and BERT.

**Strengths:**

1. Load imbalance is an important emerging topic and somewhat understudied in distributed deep learning.
2. DropCompute is a simple but appealing idea; integrating with micro-batching is a nice idea.
3. The paper includes theoretical analyses and justifications for their choices.
4. The experiments cover large, representative networks (ResNet-50, BERT) in different tasks/domains.

**Weaknesses:**

1. The motivation and significance of the paper is unclear to me: I am not sure load imbalance is an issue in practice for training. The paper gives the example of varying sentence lengths, but transformers typically concatenate sentences and pad to fixed lengths during training. Another example given is varying image sizes, but I do not think it is typical to train that way. The paper would be much stronger if there were concrete examples of widely-used and/or important use cases with load imbalance during training. This could be an issue in reinforcement learning, but that is a very different setting. Networks with conditional computation may also exhibit load imbalance, but I do not know any widely used examples.
2. A related issue to this shows up in the experimental evaluation: Delays are manually induced through simulation, rather than actual load imbalance. While this is useful for ablation studies and accuracy analysis, it reinforces my prior point.
3. There are no comparisons with other techniques to handle stragglers, such as asynchronous methods. For example, Li et al., "Breaking (Global) Barriers in Parallel Stochastic Optimization with Wait-Avoiding Group Averaging", IEEE TPDS, 2021 (see also the references therein).


**Questions:**

1. Is load imbalance an issue in practice? Can you name concrete situations, which occur in practice, where it is an issue?
2. Can you add an experiment showing DropCompute's benefit without artificially-induced delays?
3. Please add experimental comparisons with asynchronous/wait-avoiding systems (see above).
4. Why drop samples? It seems wasteful --- why not just use them in the next iteration? Is this due to a desire to follow synchronous training?
5. It might be worth clarifying that DropCompute cannot address all causes of stragglers. In particular, it seems like it does nothing for delays caused by network issues (e.g., contention, dropped packets, routing issues, etc.). This is not a criticism so much as a limitation that should be acknowledged.


**Limitations:**

Limitations are discussed.

---

> ### Author Rebuttal · Authors · 2023-08-09
>
> ### Weakness 1 response
> We thank the reviewer for the suggestion to add more concrete use case examples. Heterogeneity in computation causes issues when training in parallel schemes, and so many such workloads rely on “tricks” (such as packing of sentences and padding) in order to reduce computational variance, at the expense of performing redundant work. These approaches not only use additional resources, but also require tedious offline engineering work. Our approach doesn’t require any modification of the training task, and works “out of the box”. In addition, “waiting for the slowest worker” can also be seen as a type of “padding” and our method reduces the need for such wait. There are several examples where input shape might vary during training in vision as well as other domains (e.g., [*Tan et al. “EfficientNetV2: Smaller Models and Faster Training”, Dehghani et al.*], [*“Patch n’ Pack: NaViT, a Vision Transformer for any Aspect Ratio and Resolution”*], and [*Raffel et al. “Exploring the Limits of Transfer Learning with a Unified Text-to-Text Transformer”*]). Other reasons for compute variance are listed in (**GQ1: Is compute variance really an issue?**). We will add all those concrete examples to the paper.
>
> ### Weakness 2 response
> As written in (**GQ2: How effective is DropCompute in practice?**), we also conducted experiments on large clusters of up to 192 workers without simulative delays (section 5.2 and figure 4 and attached PDF), that prove the effectiveness of DropCompute in both gaining performance and making the system robust to stochastic non-performant outliers.
>
> ### Weakness 3 response
> We thank the reviewer for the suggested paper ― we will add it to the ‘related-work’ section.
> We focused on improving synchronous methods on a large scale, since staleness (even when bounded) is known to cause some accuracy loss of the model ― as is shown in the mentioned paper, figure 5 (-0.5% top-1 accuracy). This is also described in our paper’s “Related work ― Asynchronous optimization” paragraph. In this sense, according to table 1 in the mentioned paper, our method falls into the “No Staleness - Gradient Averaging” category, while WAGMA-SGD is a “Bounded Staleness - Model Averaging” variant. Nonetheless, in appendix A.3, we have made a comparison with Local-SGD, which is similar to WAGMA-SGD (both model-averaging with bounded staleness). Moreover, we have shown that these periodic synchronization methods are often orthogonal to DropCompute and can be integrated together while capitalizing on both (as is done in appendix A.3).
>
> ### Question 1 response
> Yes. See **GQ1: Is compute variance really an issue?**
>
> ### Question 2 response
> As written in (**GQ2: How effective is DropCompute in practice?**), we performed such an experiment in section 5.2, figure 4. In addition, we performed another experiment on a sub-optimal system which recovered ~18% performance.
>
> ### Question 3 response
>  Please refer to appendix section A.3.
>
> ### Question 4 response
> This is a great idea which we tried. However, we found that using the dropped gradients in the next iteration causes accuracy degradation similar to what is seen in asynchronous training. If the intention is to recompute the data samples in the next iterations this is also possible, as in our answer to reviewer **btsj**, in regard to his weakness (2).
>
> ### Question 5 response
> Our work focuses on solving compute variance issues, as opposed to other methods which treat communication problems, some of which are described in our “related work” section. We believe that handling straggling network connections using our approach is possible, but requires further research and HW support. We thank the reviewer, and will add ‘stragglers due to network issues’ to our limitations and future directions section.

---

> > ### Comment · Reviewer_5jtg · 2023-08-13
> > **Response**
> >
> > Thank you for taking the time to thoroughly respond to my (& others) questions and comments. This has addressed some of my concerns. I am still not fully convinced that load imbalance is a major issue in practice, but I can buy that it shows up in some situations, and definitely agree that even small performance improvements can be significant for large models (although I rather suspect that large training runs are typically executed on quiet, well-maintained systems and running such training requires significant engineering effort).
> >
> > Regarding Q4, on dropping samples: I did indeed mean to recompute the samples in some future iteration. I agree that it does not seem like it would be too hard to implement. (But it may have some interesting implications for convergence, as the order in which samples are evaluated becomes correlated with compute time; e.g., very long samples may be penalized.)
> >
> > One suggestion which occurs to me, regarding related work and compute variance, is to discuss the relation of this to classic work in systems and high-performance computing on system noise. (In my experience, on modern systems, the lessons in these works have already been applied, but it is worth at least discussing them.) For example:
> > - Petrini, Kerbyson, & Pakin, "The Case of the Missing Supercomputer Performance: Achieving Optimal Performance on the 8,192 Processors of ASCI Q", Supercomputing 2003
> > - Hoefler, Schneider, & Lumsdaine, "Characterizing the Influence of System Noise on Large-Scale Applications by Simulation", Supercomputing 2010

---

> > > ### Author Response · Authors · 2023-08-16
> > >
> > > We thank the reviewer for the positive comments on the significance of DropCompute, and for the additional feedback, including the mentioned papers on discussing system noise in HPC.

---

### Official Review · Reviewer_Zc33 · 2023-07-07

**Soundness:** 3 good
**Presentation:** 3 good
**Contribution:** 2 fair
**Rating:** 6
**Confidence:** 3

**Summary:**

This paper proposes a strategy to improve the scalability of DNN training for highly distributed workloads by allowing nodes that are lagging behind, in terms of processing time, to stop computing updates and share partial results with the other nodes in the cluster. The proposed method, DropCompute, allows the training strategy to remain synchronous while supporting scalability. The primary method to scale train while maintaining reasonable performance typically relies on asynchronous processing and/or sharing of updates between nodes in the training cluster to avoid the performance impact of straggler nodes. By maintaining the synchronous training guarantees DropCompute avoids the less desirable divergent model convergence behavior that may exist with an asynchronous approach. The authors propose a simple strategy to implement DropCompute using modern DNN software and provide theoretical and empirical results to support the claim that dynamically dropping straggler computation does not result in a negative impact on the training accuracy of the final model while simultaneously enabling faster and more reliable computation times as the number of nodes in the cluster are increased. Evaluation results consisting of runtime performance and final model training accuracy are provided to support the authors' claims regarding the effectiveness of DropCompute on real-world models.

**Strengths:**

- The authors provide a clear description and mathematical model to understand the expected behavior of dropping straggler computations. The simplicity of the method lends itself to a simple description and implementation that is clear and easy to follow.
- With the increased scale of training the influence of computational variability will continue to grow in importance. This fact leads to the natural importance of methods that are robust to possible large variations that may exist between various nodes in a cluster.
- Asynchronous training has been the go-to strategy for a number of years but the adverse impact of this strategy has been downplayed to achieve more distributed training at higher performance. DropCompute is a refreshing novel take on the distributed computation problem and is amenable to simple implementation on top of existing DNN software, which is reminiscent of dropout or dropconnect training.
- The theoretical analysis provides an interesting analysis of the expected behavior of dropcomute training and the overall impact on the final accuracy of the model.
- Evaluation data using a simulated delay environment support the performance claims presented by the authors.


**Weaknesses:**

- Though DropCompute is interesting the community of researchers with access to training environments large enough and experience sufficient computational variability to truly benefit from the approach could be small.
- How often do researchers with access to large distributed training systems experience high compute variability?
- The implementation would benefit from a lower-level implementation to increase overall efficiency. This was noted by the authors.

**Questions:**

No questions at this time.

**Limitations:**

All limitations and societal impacts were addressed by the authors.

---

> ### Author Rebuttal · Authors · 2023-08-09
>
> ### Weakness 1 response
> We appreciate the reviewer's concern regarding the limited audience that may directly benefit from DropCompute. However, the growing investment in large-scale training environments and the potential impact on efficiency justifies our focus on optimizing such settings. Also, depending on the task and system properties, medium-scale systems (16-32 workers) could also profit from DropCompute. All major cloud providers offer access to such set-ups, and many sources such as [*Falcon, W., & The PyTorch Lightning team. (2019). PyTorch Lightning (Version 1.4)], [Sylvain Gugger, B.. (2022). Accelerate: Training and inference at scale made simple, efficient and adaptable.*],and  [*NVidia Deep Learning Examples repository*] support multi-node training regimes, which indicates these are very popular.
>
> ### Weakness 2 response
> Stragglers/Compute variance can result from various reasons, and we even experienced it in some of our experiments (see **GQ1: Is compute variance really an issue?**). Researchers in particular are exposed to such cases, due to suboptimality of large research clusters. When benchmarking large scale systems (such as MLPerf), both hardware and software should be optimized. Changing the implementation, as researchers normally do, will likely destabilize the system and induce compute variance.
> Compute variance poses a significant challenge in achieving linear scaling (as shown in figure 1), and the problem increases with the scale of the system (As written in **GQ1: Is compute variance really an issue?**).
>
> ### Weakness 3 response
> We thank the reviewer for highlighting this point. This point strengthens our approach as a lower level implementation would increase the efficiency of our solution.
> A low-level implementation highly depends on framework and HW details, and we hope that with the publication of this paper, engineers will provide lower level more efficient implementation of DropCompute for their respective systems.

---

> > ### Comment · Reviewer_Zc33 · 2023-08-16
> >
> > I thank the authors for their thoughtful response to my list of weaknesses. Based on their response to my review, also considering their overall response, I have increased my score accordingly.  The contributions are interesting and worth consideration by the wider ML community of developers.

---

### Author Rebuttal · Authors · 2023-08-09

**We thank the reviewers for the insightful and constructive feedback. The evaluations provided in the paper will be improved along with the required modifications and clarifications. We will be happy for further questions and feedback.**

We thank reviewer Zc33's recognition of the novelty and importance of our work, where the reviewer described DropCompute as *“a refreshing novel take on the distributed computation problem”*. We are glad that the reviewers acknowledged the simplicity and straight-forwardness of our approach, as reviewer 5jtg wrote: *“DropCompute is a simple but appealing idea”*. We appreciate the positivity of the reviewers around our theoretical analysis which was described as *“thorough and clear”* (reviewer btsj).


## General Q&A:

**GQ1: Is compute variance really an issue?**

**A:** Yes. Compute variance can be the result of faulty hardware, clock throttling, host preemption/overhead, inefficient load balancing (for example, when training on dynamic sentence/image sizes), connectivity issues (in model parallel settings) and more.

Generally, some of these issues can be handled by non-trivial engineering work:
Faulty hardware can be tested regularly and replaced, host overhead can be avoided by latency hiding and optimization of user script, and inefficient load balancing can be handled per workload by various tricks such as sample padding and packing. However, each of these issues in itself is a single point of failure ― which, if triggered, will cause a substantial slowdown in large-scale systems.

Slowdown examples can be seen in some of our real experiments (i.e. without noise injection), which are attached in the PDF. The histograms there show real measurements in a sub-optimal system. We can clearly see a large variance in compute latency between iterations and workers. These graphs were not included in our original paper since they do not represent the final state of our system (which was regularly checked and optimized). However, as our goal is to improve robustness (i.e. performance for outlier cases), these cases are still relevant. Moreover, after the compute variance was reduced by HW and SW optimizations, we were still left with some compute variance (see Figure 2 in paper).

Although the issue of variance in sub-optimal systems is discussed intensively by system engineers, there is little reference to the matter in ML research papers. A prominent example can be seen in *Chen et. al, “Revisiting Distributed Synchronous SGD”, ICLR Workshop Track 2016*, where the performance is improved when dropping stragglers.

These examples lead us to the conclusion that, in practice, workers do not finish computation at the same time, and this can have a significant impact on the training speed. Moreover, the effect of stragglers and compute variance on the training speed is expected to get worse as the distributed scale increases. This is due to the maximal worker distribution relation stated in equation 3. For example, when modeling the additive latency as normally distributed, the average maximal worker latency increases $\\mathbb{E}[T]$ with the amount of workers $N$ as $\\mathbb{E}[T]\\sim \\text{log} N$ (the asymptotic form of eq. 4).




**GQ2: How effective is DropCompute in practice?**

**A:** The slowdown which is a result of compute variance can be avoided effortlessly using DropCompute. In a sub-optimal system with stragglers (*in attached PDF, figure (A)*) we recovered ~18% of performance. Given different systems and noise distribution, the contribution of DropCompute can be even larger (appendix section B.3). In addition, our theoretical analysis shows a greater speedup as the number of workers $N$ increases: $S_\\text{eff}\\rightarrow_{(N\\rightarrow \infty)} \infty $ (asymptotic combination of equations 4 and 6). On our own system, after the compute variance was reduced by HW and SW optimizations, we still gained a performance boost of 5% on 196 workers (section 5.2, figure 4), and the speedup is increased as the scale increases. These examples show that DropCompute makes large-scale training more robust, and “recovers” the lost performance due to any stochastic performance outlier.

Last, but not least, even a small gain to large-scale training performance can be substantial - for example, if we assume **10\\$-32\\$/hour/8xA100** (according to AWS pricing), then saving 5% of the training time for the 70B model in *Touvron et al., “Llama 2: Open Foundation and Fine-Tuned Chat Models”, 2023*, would have saved **107,50\\$-344,064\\$**, and **67,686\\$-216,598\\$** for *“BLOOM: A 176B-Parameter Open-Access Multilingual Language Model”* for the 176B model.

---

### Decision · Program_Chairs · 2023-09-21

**Decision:**

Accept (poster)

**Comment:**

The paper proposes a method for distributed synchronous training that reduces run-time variance among workers to make distributed training more reliable. The reviewers came to a consensus in favor of accepting this paper, and I agree with this consensus. I suggest that for the camera ready, the authors include some more content describing the practical effect of DropCompute on saving resources and cost in real systems.